# Stable endocytic structures navigate the complex pellicle of apicomplexan parasites

Ludek Koreny[1], Brandon N. Mercado-Saavedra [1], Christen M. Klinger[2], Konstantin Barylyuk [1], Simon Butterworth[1], Jennifer Hirst[3], Yolanda Rivera-Cuevas [4], Nathan R. Zaccai [3], Victoria J. C. Holzer[5], Andreas Klingl[5], Joel B. Dacks [2,6], Vern B. Carruthers [4], Margaret S. Robinson[3], Simon Gras[7] ✉ & Ross F. Waller [1] ✉

Apicomplexan parasites have immense impacts on humanity, but their basic cellular processes are often poorly understood. Where endocytosis occurs in these cells, how conserved this process is with other eukaryotes, and what the functions of endocytosis are across this phylum are major unanswered questions. Using the apicomplexan model *Toxoplasma*, we identified the molecular composition and behavior of unusual, fixed endocytic structures. Here, stable complexes of endocytic proteins differ markedly from the dynamic assembly/disassembly of these machineries in other eukaryotes. We identify that these endocytic structures correspond to the 'micropore' that has been observed throughout the Apicomplexa. Moreover, conserved molecular adaptation of this structure is seen in apicomplexans including the kelch-domain protein K13 that is central to malarial drug-resistance. We determine that a dominant function of endocytosis in *Toxoplasma* is plasma membrane homeostasis, rather than parasite nutrition, and that these specialized endocytic structures originated early in infrakingdom Alveolata likely in response to the complex cell pellicle that defines this medically and ecologically important ancient eukaryotic lineage.

Apicomplexa is a diverse phylum of eukaryotic intracellular parasites infecting every major animal taxon including humans. The malaria-causing *Plasmodium* spp. are responsible for over 600,000 deaths a year[1]. *Cryptosporidium* spp. are a leading cause of diarrhoea morbidity and mortality in children younger than 5 years[2,3], and *Toxoplasma gondii* is the most prevalent human parasite estimated to infect a third of the world's population[4]. While in most adults *Toxoplasma* does not cause serious illness, it can cause life-threatening congenital toxoplasmosis, fetal malformation and abortion, blindness, and encephalitis, with immunocompromised individuals most susceptible[5,6]. Other members of Apicomplexa infect economically important livestock through which they also have major impact on human wellbeing[7].

A key feature of apicomplexans is the inner membrane complex (IMC)—a near continuous array of flattened membranous vesicles (alveoli) underneath the plasma membrane and supported by a complex proteinaceous membrane skeleton. This IMC pellicle provides cell shape and strength, and is the critical platform for a gliding motility

[1]Department of Biochemistry, University of Cambridge, Cambridge CB2 1QW, UK. [2]Division of Infectious Diseases, Department of Medicine, University of Alberta, Edmonton, AB T6G 2R3, Canada. [3]Cambridge Institute for Medical Research, University of Cambridge, Cambridge CB2 1QW, UK. [4]Department of Microbiology and Immunology, University of Michigan Medical School, Ann Arbor, MI 48109, USA. [5]Plant Development, Ludwig-Maximilians-University Munich, Planegg-Martinsried 82152, Germany. [6]Institute of Parasitology, Biology Centre, Czech Academy of Sciences, České Budějovice 370 05, Czech Republic. [7]Experimental Parasitology, Department for Veterinary Sciences, Ludwig-Maximilians-University Munich, Planegg-Martinsried 82152, Germany. ✉e-mail: Simon.Gras@para.vetmed.uni-muenchen.de; rfw26@cam.ac.uk

apparatus that allows parasite tissue traversal and the mechanism for host cell invasion[8,9]. The IMC, however, separates most of the parasite's plasma membrane from its cytoplasm and is, therefore, a barrier to the material exchange processes of endocytosis and exocytosis. This is an ancient problem for these cells because the alveolae-based pellicle predates the development of apicomplexan parasitism and is a common feature shared with dinoflagellates and ciliates, the other two major lineages of the infrakingdom Alveolata[10,11]. Understanding the solutions to the challenges presented by the IMC are, therefore, equally important to understanding the major ecological functions of these organisms in ocean primary production, coral symbiosis, food webs and nutrient recycling. Adaptations for exocytosis across the IMC are best studied in apicomplexas because regulated discharge of secretory organelles micronemes and rhoptries drive the processes of host cell attachment, motile exploration, and invasion of their host cells[12–14]. This secretion occurs through the apical complex, a cytoskeletal feature integrated into the IMC that provides a window of available plasma membrane at the cell apex for vesicle docking and fusion[15]. Considerable insight into the molecular details of these exocytic processes has been achieved through decades of intense investigation[16–18], and it is evident that many of these adaptations are present in the related dinoflagellates and ciliates[19]. The details and adaptations for endocytosis, on the other hand, have received far less attention in these organisms.

The importance of endocytosis in apicomplexans has been brought into sharp focus in recent years through a link to the emergence and spread of *Plasmodium* tolerance to the lead antimalarial drug artemisinin and its derivatives (ARTs). Many ART-resistant *Plasmodium* field isolates have correlated with mutations in either a kelch (beta-propeller) domain of a protein called Kelch13 (K13) or the medium (μ) subunit of the AP-2 adaptor complex (AP-2μ)[20,21]. In the mouse model for malaria, *Plasmodium berghei*, ART-resistant parasites selected in the laboratory similarly contain mutations in AP-2μ, as well as in a deubiquitinase, UBP1[22]. While the functional significance of these mutations was initially unclear, K13, AP-2μ and UBP1, along with 11 other proteins, were recently shown to associate with the cytostome of the feeding blood-stage of *Plasmodium*[23–25]. The cytostome is a prominent membrane invagination where haemoglobin-rich host cytoplasm is taken up for digestion. Loss of function of five of these proteins at the cytostome were all seen to correlate with reduction in haemoglobin uptake, and the inactivation of eight of them conferred increased ART-tolerance. The haemoglobin degradation pathway is important for the activation of ARTs, and decreased levels of ingested haemoglobin in the mutant strains is believed to be the main cause of the resistance[24,25]. The cytostome, however, is specific to the feeding-stages of erythrocytic *Plasmodium* where the IMC is temporarily disassembled. Furthermore, of the fourteen identified K13 complex proteins in *Plasmodium*[25], only AP-2μ and Eps15 are related to previously characterized endocytic factors of other eukaryotes, although the other three canonical AP-2 subunits were enriched in AP-2μ pulldowns consistent with the tetrameric AP-2 adaptor being part of the complex[26]. The sites, machinery, or functions of endocytosis in other *Plasmodium* stages, or other apicomplexans, is largely unknown including whether they share any features of the cytostome. Small membrane invaginations named micropores have been observed throughout Alveolata by electron microscopy[27–29], but their hypothesized role in endocytosis was untested and the proteins associated with the micropore unknown.

In *Toxoplasma*, vesicular uptake from the parasite's environment has been recorded either as recycling of the surface molecule SAG1 in motile extracellular-stage tachyzoites, or uptake of proteins from the host cell cytoplasm to a lysosome-related vacuole in intracellular tachyzoites[30,31]. The sites or mechanism of uptake at the parasite plasma membrane, however, were not known and neither were the consequences of the abatement of these processes. Indeed,

the significance of endocytosis in *Toxoplasma* is as poorly understood as it is throughout the Apicomplexa. In this study we have asked where endocytosis occurs in *Toxoplasma*, what is its major function in this parasite, and does it share mechanistic features with the *Plasmodium* cytostome. We have defined a remarkably stable endocytic complex with molecular conservation to that of the *Plasmodium* cytostome as well as additional endocytic factors previously unknown in apicomplexans. This complex occurs at two to three sites that are pre-assembled and integrated into the IMC well before their endocytic functions are needed. We have developed a new endocytosis assay in *Toxoplasma* that confirms endocytic activity at these sites, and we observe that a major role for this process is maintenance of plasma membrane homoeostasis. Moreover, this complex shows conserved molecular adaptations unique to the Alveolata implying that it too represents an ancient adaptation that has contributed to the success of this medically and environmentally important lineage of eukaryotes.

## Results

### A K13-associated endocytic complex is present at the micropore of *Toxoplasma* tachyzoites

Both K13 and the canonical subunits of the AP-2 adaptor complex have orthologues throughout the Apicomplexa, but it was unknown if a molecular structure equivalent to the *Plasmodium* cytostome occurs in other apicomplexans. To test for such a structure, we determined the location and associations of these proteins associated with *Plasmodium* ART-resistance in *T. gondii* tachyzoites. We observed reporter-tagged *Tg*K13 (TGME49_262150) located at the periphery of the cell forming multiple (typically 2–3) discrete rings in the plane of the cell surface (Figs. 1A and S1). The AP-2 adaptor complex is typically comprised of four protein subunits (α, β, μ and σ) where β, in most eukaryotes, is shared with the AP-1 adaptor complex which regulates polarized sorting at the trans-Golgi network[32]. While *T. gondii* has single orthologues for the β, μ and σ subunits (TGME49_240870, TGME49_230920, TGME49_313450, respectively), we found two unusually divergent proteins that shared similarity to the canonical α subunit. One protein (designated here *Tg*AP-2α, TGME49_221940) shares similar size and strong sequence identity to the canonical AP-2α proteins but has an unusually divergent C-terminal 'ear domain' that typically mediates cognate AP-2 interactions with other proteins. The second *T. gondii* protein contains the highly conserved C-terminal ear domain but the rest of the protein lacks similarity to AP-2α. We designate the second protein KAE (K13 complex-associated, alpha-ear containing protein, TGME49_272600). All five of these AP-2-related proteins co-locate with *Tg*K13, with the β subunit displaying additional location within the cell consistent with a shared role with AP-1 (Figs. 1A, S1 and S2A)[32]. The AP-2 components locate within the lumen of the K13 ring presenting as a tapered funnel in cross-section by 3D-SIM super-resolution microscopy (Fig. S2A). These data are consistent with endocytic pits lined by AP-2 at the *T. gondii* plasma membrane within a K13-associated ring.

Dynamin typically mediates vesicle fission during endocytosis. *T. gondii* has three dynamin-related proteins (Drp) with DrpA and DrpB known to have roles in apicoplast division and vesicle trafficking to rhoptries and micronemes, respectively[33,34]. DrpC (TGME49_270690) function is less well understood although its depletion leads to failure of cell division and proliferation[35]. DrpC pulldowns have implicated an association with both AP-2 components and K13[36] and, when we C-terminally reporter tagged this protein, DrpC is seen associated with the K13 complex (Figs. 1B, S1). DrpC is located at a relatively interior position with respect to K13 and appears in one of two forms − a ring of similar diameter to that of K13, or a conical punctum of smaller diameter−consistent with cycles of constriction (Figs. 1B and S2C–F).

The sites of endocytosis at the apicomplexan cell surface have been hypothesized to be ultrastructural features termed 'micropores'. Micropores are small extensions of the plasma membrane that insert in

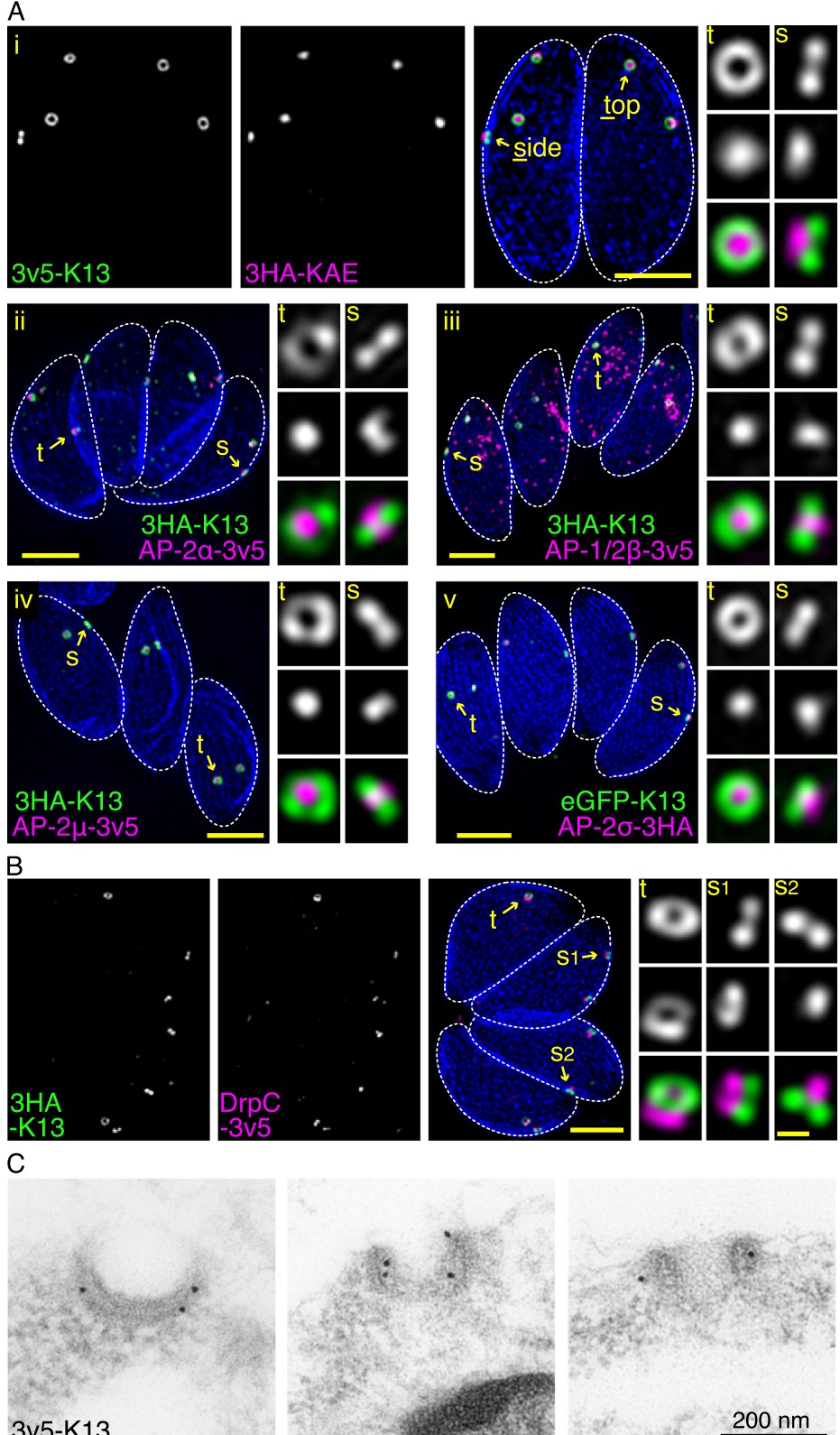

**Fig. 1 | Endocytosis-related AP-2 and DrpC associate with K13 at peripheral micropore structures in *T. gondii*. A** Collapsed projections of 3D-SIM images of intracellular parasites within the host cell vacuole showing K13 (green) with five different AP-2 related proteins: (i) KAE, (ii) AP-2α, (iii) AP-2μ, (iv) AP-1/2β and (v) AP-2σ (all magenta). **B** K13 (green) with dynamin related protein DrpC (magenta). The IMC1 (blue) shows the parasite inner membrane complex, and zoomed panels show micropores either in side (s) or top (t) projections as indicated. Reporter epitopes and the terminus of fusion are shown in the figures. See Fig. S1 for wide-field fluorescence images and Fig. S2 for further examples. Scale bars for large and small panels are 2 μm and 0.2 μm, respectively. **C** Immuno-TEM against v5-tagged K13 showing gold labelling specific to the classic micropore structure. See Fig. S3 for further cell examples.

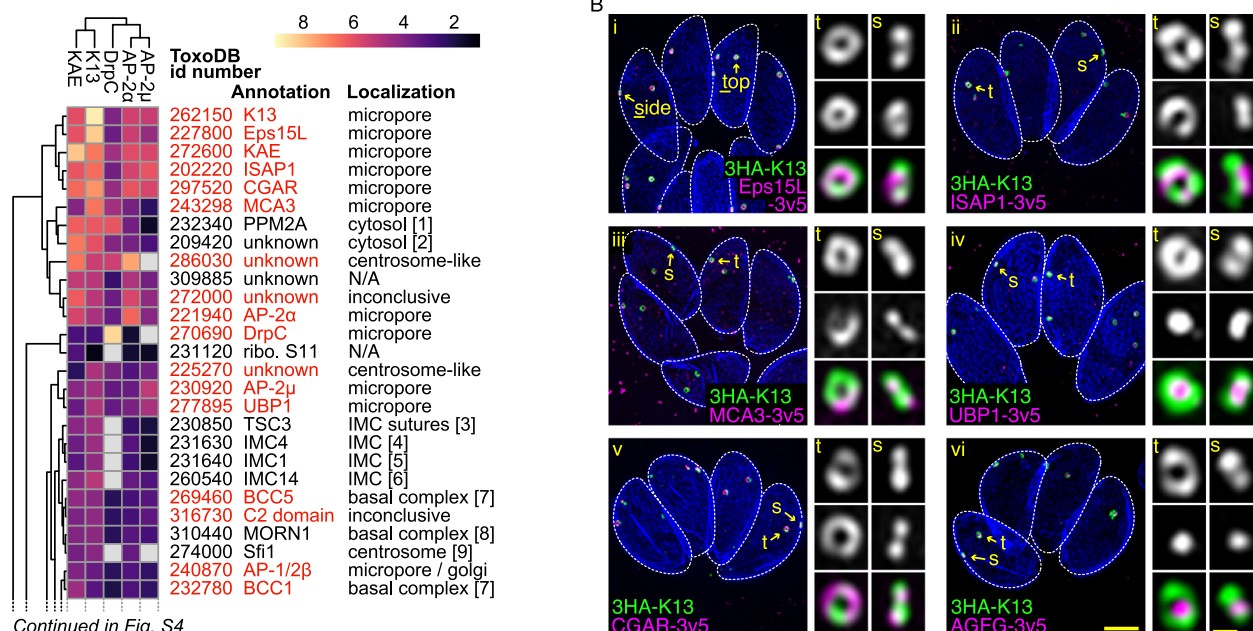

**Fig. 2 | *Toxoplasma* micropore shows conservation with the *Plasmodium* cytostome but also possesses unique proteins. A** Heat map of BioID-enriched proteins determined with five bait proteins. Results are clustered according to similarity of results between baits and ordered by fold enrichment (colours). Proteins reporter tagged in this study are shown in red, and inferred cell location given. Numbers in brackets refer to previously published localizations: [1][90], [2][91], [3][42], [4][92], [5][93], [6][94], [7][95], [8][96], [9][97]. Full heat map is shown in Fig. S4A. **B** Collapsed projections of 3D-SIM images of intracellular parasites showing K13 (green) with (i) Eps15L, (ii) ISAP1, (iii) MCA3, (iv) UBP1, (v) CGAR and (vi) AGFG (all magenta). IMC1 or GAP45 (blue) mark the parasite pellicle, and zoomed panels show micropores either in side (s) or top (t) projections as indicated. Reporter epitopes and the terminus of fusion are shown in the figures. Scale bars for large and small panels are 2 μm and 0.2 μm, respectively.

a break in the IMC and that are outlined by two circumferential rings of electron-opaque material[27]. To test if the K13 complex occurs at the micropores of *Toxoplasma* we immuno-gold-labelled K13 epitope-tagged cells and imaged thin sections of parasites by transmission electron microscopy (TEM). The K13 labelling decorated a surface structure with all the features of the micropore supporting that this is an endocytic structure in apicomplexans (Figs. 1C and S3).

To identify further molecular components of the K13 complex-associated micropore in *T. gondii*, we used proximity-dependent biotinylation (BioID) with five bait proteins (K13, KAE, AP-2α, AP-2μ, DrpC). Enrichment of biotinylated proteins (≥2-fold) for each bait compared to a *birA*\*-negative control cell line identified proteins as possible K13 complex interactors, with many candidates identified by multiple of the baits (Figs. 2A, S4A and Supplementary data 1–2). Several known proteins of the general IMC were identified in the enriched set consistent with the peripheral location of this structure. Twenty proteins of previously uncharacterized location were reporter tagged (Figs. 2A and S4) and this validated a further five proteins that exclusively located at the K13 complex (Figs. 2B and S4B). Four of these proteins are orthologues of *Plasmodium* cytostome proteins[25]: Eps15-like (Eps15L, TGME49_227800) that contains two Eps15-homology (EH) domains and a Coiled-coil region; UBP1 (TGME49_277895), a putative deubiquitinase that is also implicated in ART-tolerance; metacaspase MCA3 (TGME49_243298); and a Coiled-coil and Gas2-related (GAR) domain protein (CGAR [TGME49_297520], previously named proteophosphoglycan [PPG] 1 in *T. gondii*). The fifth protein, ISAP1 (TGME49_202220), is specific to coccidian apicomplexans and consists of coiled-coil domains but no other conserved features[37]. Given the significant overlap of *T. gondii* K13 complex proteins with the *Plasmodium* cytostome we also tested two further orthologues, Kelch13 interaction candidate (KIC) 3 and KIC7, that were identified by BioID of the cytostome although not detected in our BioID results[25]. KIC7 shares similarity with AGFG

(Arf-GAP domain and FG repeat-containing protein) which, in other organisms, has been implicated with Eps15 interaction and endocytosis[38,39]. The *T. gondii* AGFG orthologue (*Tg*AGFG, TGME49_257070) was exclusively located with the *T. gondii* K13 complex (Figs. 2B and S4B). The orthologue of KIC3 (TGME49_265420), however, does not localize to the K13 foci in *T. gondii* (Fig. S4B). The reporters for Eps15L, MCA3, CGAR and ISAP1 collocate with K13 as a ring, and UBP1 and AGFG both form a punctum within the K13 ring. These six further K13 complex proteins support that a similar molecular apparatus to the *Plasmodium* cytostome occurs at the peripheral micropores of *T. gondii* tachyzoites.

## The K13 complex is a stable and essential component of the *Toxoplasma* inner membrane complex

The *Plasmodium* blood-stage cytostome occurs in the trophozoite stage where the inner membrane complex (IMC) is absent and, hence, access to the parasite plasma membrane is unencumbered. The plasma membrane of *T. gondii* tachyzoites, on the other hand, is heavily fortified by a robust IMC comprising a dense filamentous proteinaceous network that supports rectangular membranous cisternae sutured together as a quilt adjacent to the cytosolic face of the plasma membrane[40,41]. To determine how the K13 complex is positioned with respect to this IMC, and how it might have access to the plasma membrane for possible endocytic function, we co-labelled several components of these peripheral structures with *Tg*K13.

The *Toxoplasma* IMC membrane plate boundaries are demarcated by suture protein ISC3[42]. K13 is consistently located at these boundaries (Fig. 3A). K13 occurs either within the longitudinal or the transverse plate boundaries, including the most apical boundary at the base of the conically shaped apical cap plate. While predominantly in the apical half of the cell, the K13 rings are seemingly haphazardly dispersed in IMC position. ISAP1 was independently located to puncta

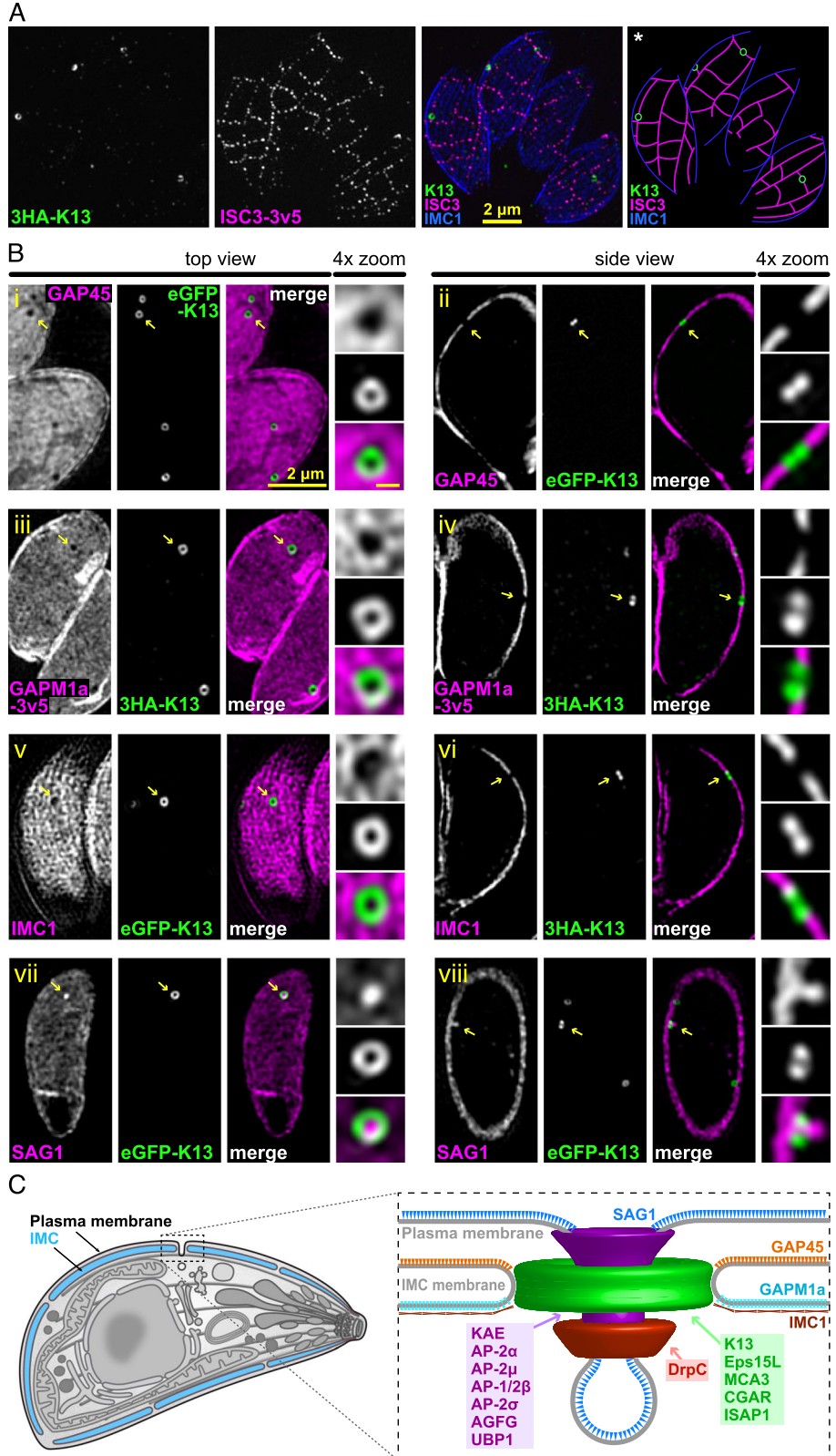

**Fig. 3 | The micropore occurs in openings of the IMC at the alveolar plate boundaries. A** 3D-SIM images of intracellular parasites showing K13 (green) with a suture protein ISC3 (magenta). *The suture lines were extrapolated in the rightmost image based on the ISC3 signal. **B** Top and side 3D-SIM projections of micropores with markers for the IMC outer and inner membrane proteins GAP45 (i–ii) and GAPM1a (iii–iv), respectively; IMC subpellicular network protein IMC1 (v–vi); and GPI-anchored surface protein SAG1 (vii–viii) (all in magenta). Yellow arrows correspond to structures shown in the magnified panels, and all scale bars are 2 μm and 0.2 μm for the overview and zoomed images, respectively. **C** Model of the endocytic K13 micropore complex in the pellicle of *T. gondii* tachyzoite from these data and Figs. 1 and 2. IMC inner membrane complex. See Figs. S5 and S6 for further examples.

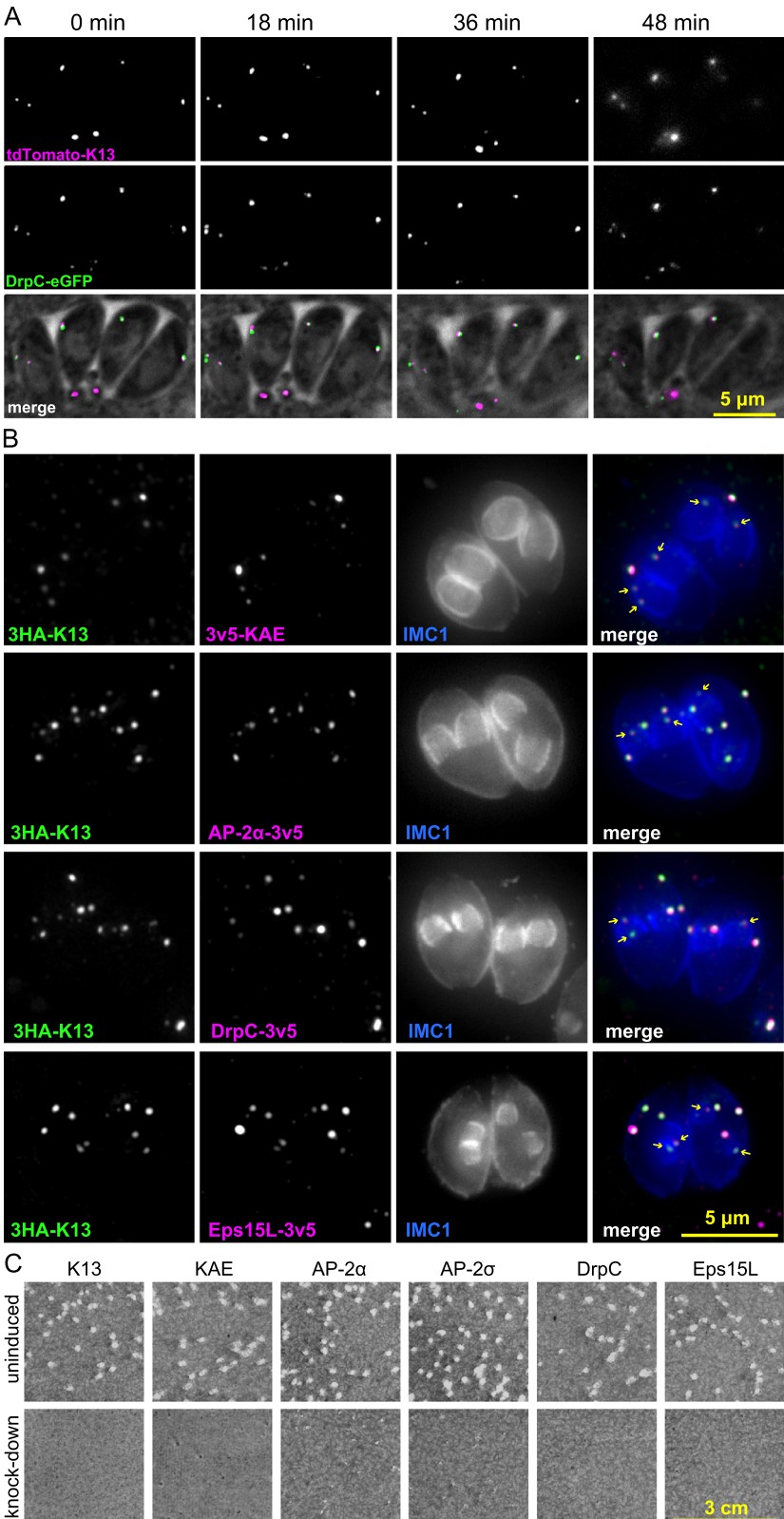

**Fig. 4 | The K13 complex is stable over time, is assembled early in IMC formation during cell replication, and is required for parasite lytic cycle progression.** **A** Live-cell time-lapse sequence of four parasites in a host vacuole showing that K13 and DrpC locations are stable over time. **B** Wide-field fluorescence images of cells fixed during the formation of daughter cells that are evident by internal IMC1 (blue)

pellicle 'cups'. K13 (green) is co-stained with KAE, AP-2α, DrpC and Eps15L (all magenta) with weaker protein signals present for all in the developing daughters (arrows), some of which are seen at the pellicle leading edges. **C** Seven-day plaque growth assays in six cell lines with K13 complex gene expression individually suppressed by ATc. Representative results of three independent assays are shown.

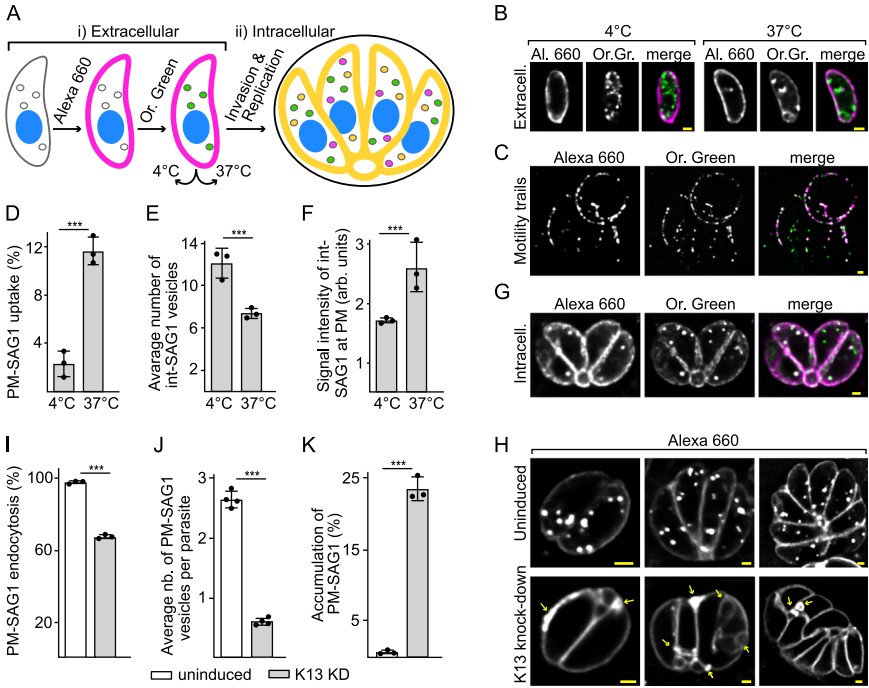

**Fig. 5 | SAG1-Halo-bound ligands report on endocytosis which is reduced in K13-depleted parasites. A** Schematic of differential labelling of SAG1 pools at the plasma membrane surface (PM-SAG1 stained with Alexa 660, magenta) or in internal vesicles (Int-SAG1 stained with Oregon Green, green) and tests for endocytic recycling at either non-permissive (4 °C) or permissive (37 °C) temperatures (i). Infection of hosts after labelling allows these processes to be monitored during intracellular growth (ii) (orange indicates co-location of signals). **B** Ligand detection of labelled extracellular cells incubated at either 4 °C or 37 °C. **C** Motility trails of labelled SAG1 left by gliding parasites at 37 °C. Differences according to temperature treatment in: **D** the percentage of cells with internalized PM-SAG1 as a measure of endocytosis (*P* value = 3.52E−5), **E** the number of Int-SAG1 vesicles per cell as a measure of exocytosis (*P* value = 3.03E−16), and **F** the intensity of Int-SAG1 at the cell surface as a further measure of exocytosis (*P* value = 9.82E−6). **G** Redistribution

of differentially labelled extracellular cells (PM-SAG1 and Int-SAG1) viewed 24 h after invasion and intracellular growth. **H** Alexa 660-labelled PM-SAG1 after 24 h of growth without or with K13 depletion. Reduced uptake of PM-SAG1 to intracellular vesicles is quantified as **I** percentage of parasites with internalized PM-SAG1 (*P* value = 7.75E−16) and **J** the average number of PM-SAG1-positive vesicles per cell (*P* value = 1.13E−12). K13-depleted cells showed accumulations of extra PM-SAG1-positive membranes (arrows in H) which occurred in a higher percentage of cells than for the uninduced controls (**K**) (*P* value = 1.77E−9). Three biological replicates were used for all analyses; All *P* values are 0 ≤ *P* ≤ 0.001, ***, error bars are standard deviations and the centre measurement of the graph bars is mean. Two-sided Student's *T*-test was used for all the comparisons with no adjustments. All scale bars = 1 μm. Source data are provided as a Source Data file.

within the IMC sutures and collocated with Eps15L, KAE, CGAR and DrpC[37], further supporting this position of the micropores between IMC plates. Markers for the IMC cisternae outer (GAP45) and inner (GAPM1a) membranes[43] showed that the K13 ring is positioned in windows where these membrane markers are excluded (Figs. 3Bi–iv and S5). Such mutual exclusion is also seen for K13 and the proteinaceous network marker IMC1 (Figs. 3Bv–vi and S6A)[40]. Together these data indicate coordinated interruption of the IMC where the K13 complex provides an interface between the plasma membrane and the cell cytoplasm. The outer leaflet of the plasma membrane is dominated by the GPI-anchored surface antigen glycoprotein SAG1. When SAG1 is immuno-labelled a concentration of SAG1 is frequently seen at the K13 sites where SAG1 penetrates the ring in a finger-like depression from the cell surface (Figs. 3Bvii–viii and S6B). The arrangements of these IMC and plasma membrane markers with the K13 rings are all consistent with the micropore ultrastructure as a surface depression between IMC plates (Fig. 3C)[27].

Endocytosis is typically a dynamic process with many of the molecules that drive it recruited then released after endocytic events and in response to multiple signalling networks[44–46]. To test if the K13 endocytic complex shows this typical dynamism, we monitored the locations of K13 and DrpC over time using live-cell fluorescence microscopy. We saw no evidence of temporary recruitment or release of these proteins, and the number and relative positions of the complexes was unchanged over at least 48 min (Fig. 4A). These time-lapse data suggest stable structures permanently embedded

within the IMC pellicle, and this raised the question of when these structures are first assembled. During the division of most apicomplexan cell stages the daughter cell IMC is prefabricated as internal cups within the mother cell[40,47]. These IMC cups consist of both the membranous cisternae and proteinaceous network elements, and they develop to a state of relative maturity before organelles are sorted into the base of each one. Only during cytokinesis is the mother cell's plasma membrane recruited to each emerging daughter. To determine how the development of the IMC accommodates the K13 complex, we examined the location of K13 complex components during the early assembly of daughter cells. Invariably, we saw K13, KAE, AP-2α, EPS15L and DrpC present as the characteristic complexes in these developing daughter IMCs (Fig. 4B). In some images K13 complex components are seen recruited at the growing margin of the IMC1 cup suggesting that this complex is part of the initial assembly of the IMC structure. It is note-worthy that these proteins are all recruited before the plasma membrane is associated with the IMC that ultimately occurs with daughter cell emergence.

To test if the components of the K13 complex are necessary for tachyzoite growth, seven-day plaque growth assays were performed in host cell monolayers in cell lines engineered for inducible protein knockdowns using an ATc-responsive promoter. We targeted six proteins spanning the different functional components of the complex for individual depletion (K13, KAE, AP-2α, AP-2σ, DrpC and Eps15L) and all mutants showed strong growth phenotypes (Fig. 4C).

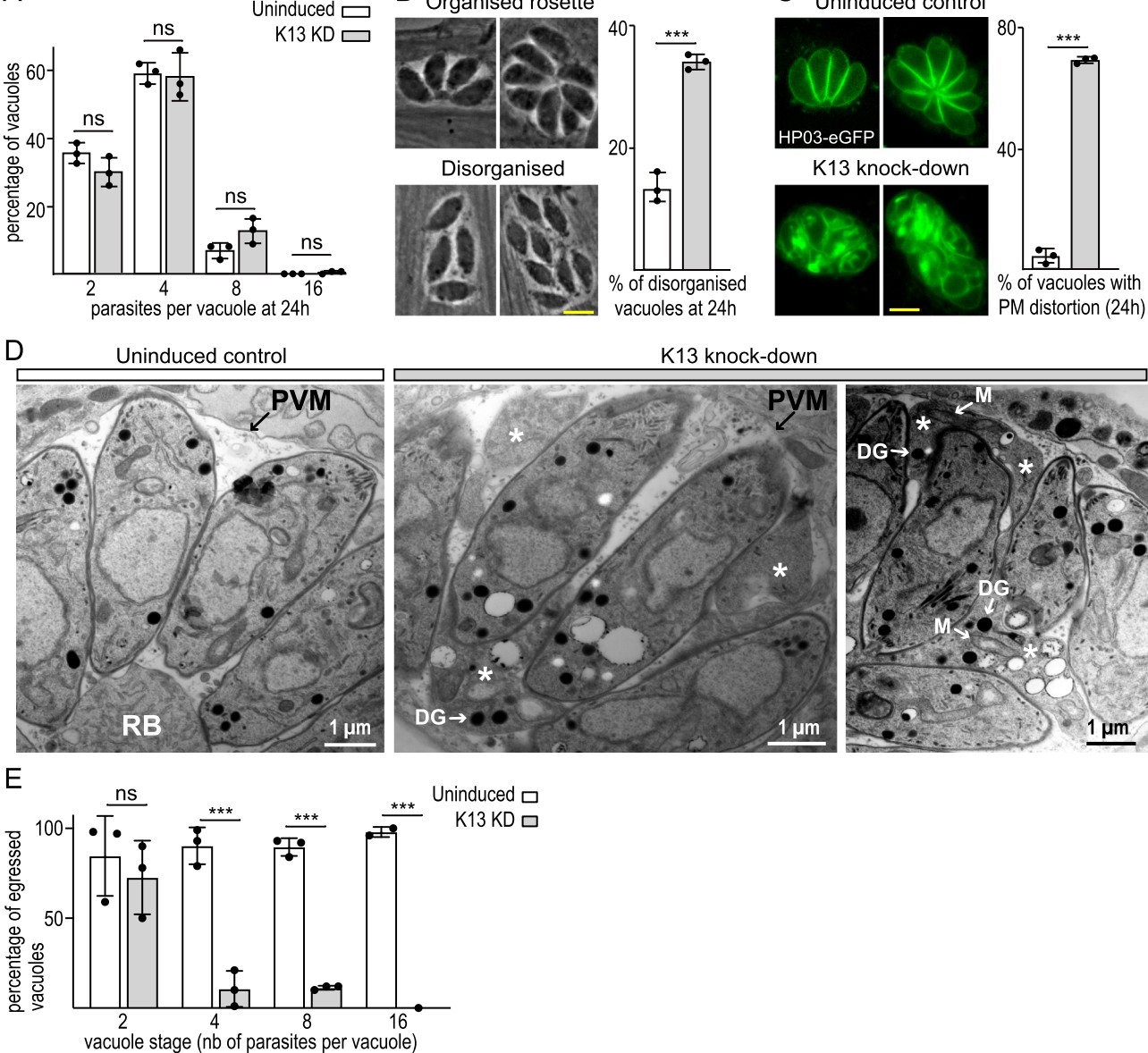

**Fig. 6 | K13-depletion disrupts parasite order, integrity and egress but not replication rate. A** Number of parasites per parasitophorous vacuole after 24 h of post invasion replication with or without 72 h of ATc-induced K13 depletion; *P* values are 0.1401 (2 parasites), 0.8919 (4 parasites), 0.0688 (8 parasites) and 0.1161 (16 parasites). **B** Percentage of disorganized vacuoles after the same treatments as (**A**). Disordered vacuoles are scored as those that lack ≥75% of parasites sharing a common posterior orientation as shown by four- and eight-cell parasite rosettes (examples imaged by phase contrast); *P* value = 0.0002. **C** Live cells expressing a plasma membrane marker HP03-eGFP after the same treatments as (**A**) and (**B**) showing increasing lack of organization and PM integrity with K13-depletion (see Fig. S8 for more examples); *P* value = 0.00001. The scale bars in (**B**) and (**C**) are 5 μm. **D** TEM images of parasites within the parasitophorous vacuole with K13 either

present or depleted. With K13 depletion the spaces between parasites are filled with parasite cytoplasm bounded by a single membrane (asterisks) and containing organelles including mitochondria (M) and dense granules (DG). RB, residual body; PVM, parasitophorous vacuole membrane. **E** Egress assay showing percentage Ca²⁺ ionophore-induced egressed vacuoles according to vacuole parasite number for K13-depleted versus control cells; *P* values are 0.5299 (2 parasites), 0.0007 (4 parasites), 0.0000 (8 parasites). Three biological replicates were used for all analyses; *P* values are indicated as 0.05 < *P* ≤ 1, ns; 0 ≤ *P* ≤ 0.001, ***, error bars are standard deviations and the centre measurement of the graph bars is mean. Two-sided Student's *T*-test was used for all the comparisons with no adjustments. Source data are provided as a Source Data file.

## Development of a new endocytosis assay for intracellular *Toxoplasma* tachyzoites

Given that the composition of the *T. gondii* K13 complex includes several endocytosis-related proteins, we sought to verify that endocytosis is associated with this structure in *T. gondii* tachyzoites. It has been previously shown that fluorescent reporter proteins expressed in the cytoplasm of parasitized host cells can accumulate in the *T. gondii* 'plant-like vacuole' or VAC[30]. These signals are only seen when proteolytic digestion by the VAC cathepsin L is inhibited, consistent with

this compartment functioning as a terminal lysosome. Although insight was recently gained for how host proteins traverse the parasitophorous vacuole membrane in which the parasites reside[48], how the material crosses the parasite plasma membrane is uncharacterized. Nevertheless, internalization presumably involves an endocytic process at the parasite surface, and this system offered the opportunity to assay for this activity. We tested if K13 depletion in tachyzoites led to a detectable difference in VAC-accumulation of host reporter protein. K13 was depleted first for 48 h in wildtype host cells (Fig. S7A), and

these cells were then allowed to invade host cells expressing cytosolic mCherry. Here they were allowed to grow for 24 h with continued K13 suppression and cathepsin L inhibition. This assay detected no difference between K13 depletion and controls (Fig. S7B). However, in either treatment only approximately 5% of parasites showed detectable accumulation of mCherry indicating that an alternative assay was required to better discern any effects on endocytosis in the knockdowns.

The *T. gondii* surface protein SAG1 is taken up through endocytosis in extracellular tachyzoites[31] so we targeted this molecule to develop a new endocytosis assay for parasites within the host cell vacuole. Previous assays for endocytosis of SAG1 in extracellular parasites used antibodies targeting SAG1[31] but these antibodies are known to also inhibit invasion[49] and, therefore, are incompatible with the analysis of intracellular parasites. Instead, we modified the *sag1* gene to express a fused HaloTag that can catalyze covalent linkage to exogenous fluorescent ligands[50]. We used two ligands, one membrane impermeable (HaloTag-Alexa Fluor-660 ligand) and one membrane permeable (HaloTag-Oregon Green) to differentially label the external pool of SAG1 (at the plasma membrane, PM-SAG1) from internal pools of SAG1 (internal SAG1-containing vesicles, Int-SAG1) (Fig. 5A). Application of the membrane impermeable ligand first was shown to saturate binding to the PM-SAG1 pool such that secondary labelling with membrane permeable ligands exclusively labelled the Int-SAG1 pools (Fig. S7C–F).

To test if this labelling strategy can detect the endocytic/exocytic cycling of SAG1, we first tested extracellular tachyzoites where uptake of SAG1 was initially reported[31]. PM- and Int-SAG1 pools were separately labelled and then parasites were incubated for 60 min at temperatures either permissive (37 °C) or restrictive (4 °C) to vesicular transport (Fig. 5A). At 37 °C internalization of PM-SAG1 (Alexa Fluor-660-labelled) into cytosolic puncta was observed and, as evidence of this being vesicle-mediated, this was significantly higher than for the 4 °C treatment (Fig. 5B, D). Concurrently, Int-SAG1 (Oregon Green-labelled) was observed redistributed at the cell surface in a temperature-dependent manner and this ligand was also detected in the parasite motility trails that are known to contain sloughed surface SAG1 (Fig. 5B, C, F). Furthermore, fewer Int-SAG1 vesicles were observed per parasite in the 37 °C treatment (Fig. 5B, E). These data are consistent with both endocytic and exocytic cycling of SAG1 being detectable using ligand-conjugated SAG1-Halo which is equivalent to that seen using antibodies[31].

To evaluate endocytic/exocytic activity of intracellular parasites, SAG1 dual stained parasites (PM-SAG1 Alexa 660; Int-SAG1 Oregon Green) were allowed to invade host cells (Fig. 5A). After 24 h of growth and replication, the PM-SAG1 signal was observed in internal vesicles in the parasites of most vacuoles (98.3 ± 1.6%) and the Int-SAG1 signal was detected at the parasite plasma membrane in all (Fig. 5G). These signals show that PM-SAG1 is endocytosed and apparently recycled in intracellular tachyzoites, and that SAG1 from the initial mother cell continues to be recycled to daughter parasites through subsequent cell divisions. SAG1, therefore, provides a very effective marker of endocytosis in intracellular tachyzoites, as well as revealing the exocytosis and recycling of surface molecules in intracellular parasites including to daughter cells.

### K13 depletion disrupts endocytosis and plasma membrane homoeostasis in *Toxoplasma*

To test if the K13 complex is involved in endocytosis, the SAG1-endocytosis assay was employed in K13-depleted cells (iΔHA-*Tg*K13-SAG1-Halo). These parasites were pre-treated with ATc for 48 h to deplete detectible K13. Parasites were then egressed and PM-SAG1 was labelled with the Alexa 660 ligand before being allowed to reinvade host cells and replicate for a further 24 h with continued ATc-suppression of K13. Endocytic activity was then determined both by

the number of parasites with endocytosed PM-SAG1 vesicles and the average number of these vesicles per parasite. Compared to the controls, both measures of endocytosis were significantly reduced in the K13-depleted cells: 97.2 ± 1.3% versus 66.8 ± 1.2%, and 2.6 versus 0.6 vesicles, respectively (Fig. 5H–J). We noted also that the K13-depleted cells showed accumulation of extra PM-SAG1-positive membranes attached to or between the parasites and these irregularities were significantly different to the controls (Fig. 5H, K). These data support that K13 is associated with the sites and activity of endocytic recycling of SAG1 in intracellular *T. gondii* tachyzoites.

*Plasmodium* blood-stage parasites are dependent on cytostome-mediated endocytosis of erythrocyte cytoplasm for its nutrition, however, the roles for endocytosis in *T. gondii* intracellular tachyzoites are poorly understood. If endocytosis is necessary for nutrition and growth, a retardation in cell replication would be expected when this process is disrupted. To test for an effect on replication, we depleted K13 for 48 h in host cells, and then inoculated these parasites into new host cells with ongoing K13 suppression. No difference was seen in parasite replication rate over 24 h of further growth in the absence of K13 (Fig. 6A). The organization of the parasites within the parasitophorous vacuole, however, was noted to be atypical. Replicating *T. gondii* tachyzoites are tightly organized and typically present as a rosette in in vitro culture in fibroblast monolayers (Fig. 6B). These rosettes form by parasites being tethered at their bases to the so-called 'residual body'—a membrane-bound structure that connects and physically coordinates replicated cells in a state of incomplete cytokinesis until egress is triggered[51]. At 24 h post reinfection, the K13-depleted vacuoles showed an increased frequency of disorder compared to K13-replete cells (Fig. 6B), and with further time individual parasites were no longer clearly distinguishable in the vacuoles by transmitted light microscopy (Fig. S8). We, therefore, used a live-cell plasma membrane marker, a GFP fusion of the integral membrane carrier protein HP03, to visualize the parasite bounding membrane during K13-depleted parasite development[52]. HP03 location showed an accumulation of excess membrane including extra signals at the base and between the parasites (Figs. 6C and S8). TEM of thin sections of parasite vacuoles showed that, rather than parasites being organized around and connected to a discrete residue body (uninduced control), K13-knockdown parasite vacuoles showed inter-parasite spaces filled by single-membrane-bound parasite cytoplasm (Fig. 6D). These extensions lacked an IMC but often contained recognizable parasite organelles. These extensions are consistent with the accumulation of Halo-tagged SAG1 and HP03-GFP seen between parasites (Figs. 5H, 6C and S8). To test if these K13-deficiency phenotypes occur with depletion of other endocytic proteins at the micropore, we also examined parasite replication rate, vacuole organization, and parasite ultrastructure after DrpC depletion. The same phenotypes of normal replication rate, but loss of vacuole organization and excessive plasma membrane extensions were seen for DrpC depletion as for K13 depletion (Fig. S9). Collectively, these data suggest that a failure of plasma membrane reuptake when endocytosis is inhibited leads to an excess of membrane and breakdown of the organization of tachyzoites in the parasitophorous vacuole.

The plaque assay phenotypes of inhibited lytic cycle progression when K13 complex proteins were depleted are in apparent conflict with normal parasite replication rates, at least in these early stages after protein depletion. We, therefore, asked if endocytosis inhibition, which results in the disruption of plasma membrane homoeostasis, might affect parasite egress which is necessary for the subsequent host cell infections and plaque development. With K13 depleted as for the replication assay, we tested egress competency induced by a Ca²⁺ ionophore, scoring vacuoles according to their parasite number prior to ionophore treatment. Vacuoles with two parasites egressed with equivalent high rates of efficiency to controls, however vacuoles with 4, 8 or 16 parasites showed very low rates of egress competence

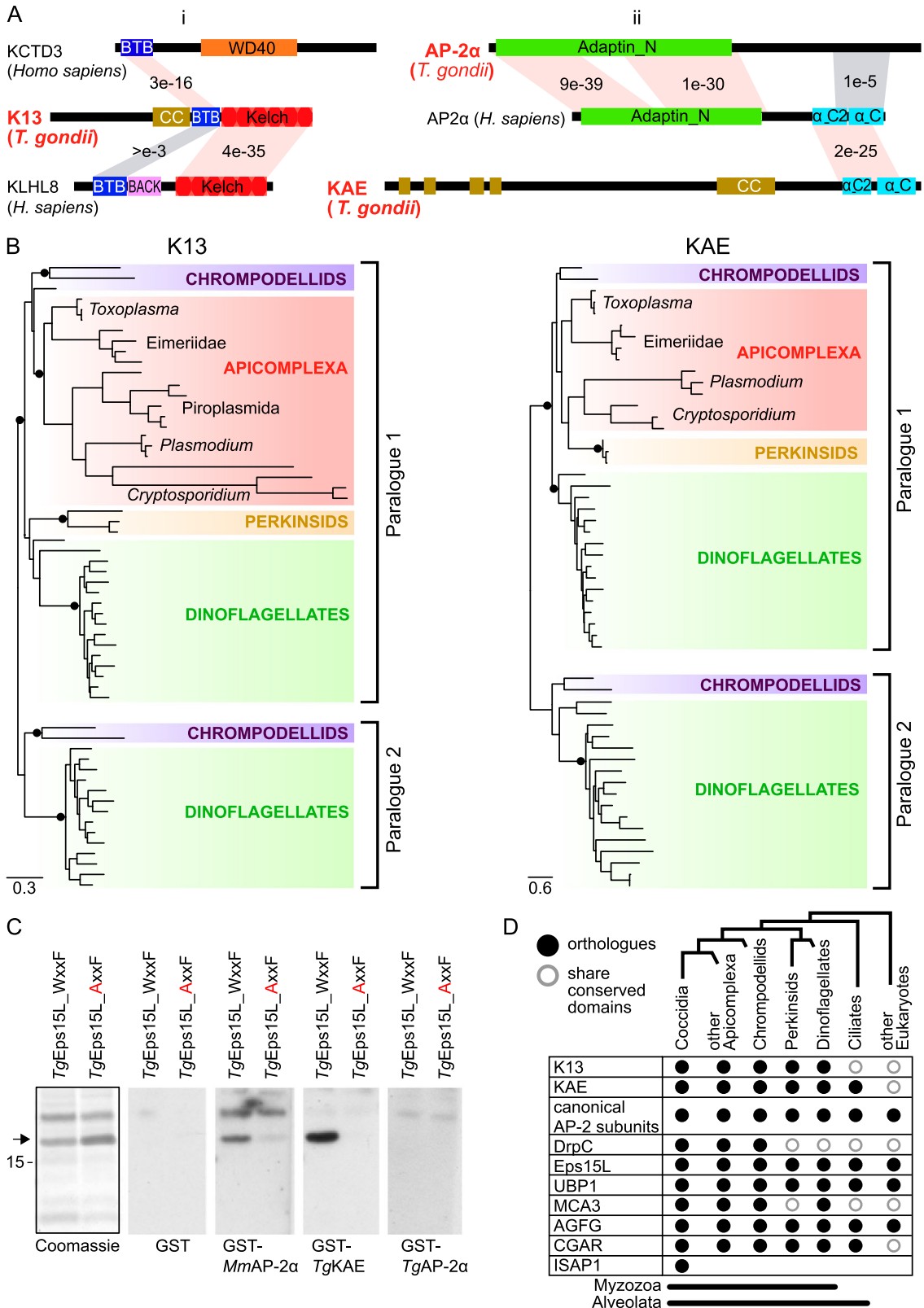

**Fig. 7 | K13 and AP-2 unique traits co-evolved in Myzozoa. A** Domain architectures of: (i) K13 compared to proteins from humans, which demonstrate high sequence similarity to the BTB and Kelch domains, and (ii) apicomplexan AP-2α and KAE compared to human AP-2α as canonical representative. BLASTP *E*-values indicate relative conservation between common domains. Conserved domains shown in colour; CC, coiled coil. **B** Maximum-likelihood phylogenies for K13 and KAE. SH-like aLRT branch supports over 0.9 are indicated by black dots, and complete phylogenies are shown in Fig. S11D. **C** Far-Western blots with immobilized fragments of *Tg*Eps15L (arrow) containing either the native (WxxF) or mutated (AxxF) of the predicted AP-2 ear-binding domain (Coomassie-stained gel, black outline). GST-fused ear domains of AP-2 candidate proteins from *Mus musculus* (*Mm*) or *T. gondii* (*Tg*) (or GST alone as negative control) were allowed to bind to the Esp15L fragments and visualized by anti-GST staining. The marker size is 15 kDa. Source data are provided as a Source Data file. **D** Distribution of K13 complex proteins in myzozoan and related lineages.

(Fig. 6E) which provides an obvious mechanism for the plaque assay result.

## The K13 complex is an ancient endocytic adaptation common to Myzozoa

Given the common presence of a specialized K13-associated endocytic structure in both *Plasmodium* and *Toxoplasma*, we asked how ancient this association is and in what other taxa it might be found. K13, as signature for this structure, is comprised of a highly conserved coiled-coil region and BTB domain attached to the β-propeller kelch domain (Fig. 7A). While these domains contribute to a wide range of eukaryotic proteins, the architecture of the K13 protein, including the conserved coiled-coil sequence, could only be found within apicomplexans and other members of the Myzozoa (chrompodellids, dinoflagellates, perkinsids). Both dinoflagellates and chrompodellids typically have two K13 proteins and the molecular phylogeny of K13 shows that an ancestral gene duplication gave rise to these two paralogues but that apicomplexans lost one of these (Fig. 7B). A further conspicuous feature of the K13 complex is the duplication of the AP-2α ear domain to form two proteins—*Tg*AP-2α with a degenerate ear domain, and KAE that contains a conserved C-terminal ear domain but this larger protein otherwise lacks further AP-2α similarity or other conserved domains (Figs. 7A and S10). Structural modelling of the ear domain of KAE predicts that it maintains the conserved motif-binding domains for its interaction partners including the sandwich domain that binds WxxF motifs found in proteins such as Eps15 (Fig. S11A–C). *Tg*Eps15L is part of the K13 complex (Fig. 2), and we verified the predicted interaction by far-western blot interaction assays where the ear domain of KAE bound *Tg*Eps15L in a WxxF motif-dependent manner (Fig. 7C). *Tg*AP-2α did not bind *Tg*Eps15L in this assay (whereas the murine AP-2α did) providing strong evidence of a bifurcation of canonical AP-2α function in this endocytic complex. We find the KAE protein throughout myzozoan organisms where K13 is present. Furthermore, an ancient duplication creating two KAE paralogues was maintained only in dinoflagellates and chrompodellids, and this correlates with the duplication of K13 and its subsequent reduction to one paralogue in apicomplexans (Figs. 7B, S10 and S11D). This evolutionary pattern suggests cognate interactions between K13 and KAE. Collectively, these molecular signatures of the K13 complex found throughout Myzozoa indicate common adaptations of endocytosis that evolved early in the evolution of this group and have been maintained throughout its extant diverse members.

## Discussion

The recent discovery that endocytosis at the cytostome of *Plasmodium* blood-stage parasites is central to resistance mechanisms against the lead antimalarial artemisinin has focused new attention on this process in *Plasmodium*[23–25]. However, it also raised important broader questions in apicomplexan biology. What is the origin of the cytostome structure in *Plasmodium*? Do homologous structures occur in other stages or other apicomplexans? Indeed, what is the site of endocytosis in other pathogen models such as *Toxoplasma*, what are the properties of such sites, and what is the dominant role for endocytosis in this system? While the mechanisms of endocytosis have been extensively studied in select eukaryotic models (e.g., mammals and plants) these processes are poorly understood in other important organisms including apicomplexans.

As evidence of the *Plasmodium* cytostome being derived from a common structure, we have identified the sites of endocytosis in *T. gondii* which share at least 11 proteins in common with the cytostome. Five of these are classically associated with endocytosis: Eps15L and the four AP-2 adaptor complex components. Additionally, the Arf-GAP-related AGFG (KIC7 in *Plasmodium*) is implicated in endocytosis in humans where it contains NPF motifs that bind to the EH domains of Eps15[39], and these NPF motifs also occur in the apicomplexan AGFG.

Other components of the endocytic complex found in *Toxoplasma* and *Plasmodium* do not have direct orthologues in characterized endocytic machineries but do share potential activities with such systems. KAE, with the conserved α ear domain, likely maintains some of the interactions of the AP-2 complex with other endocytic factors. In mammals, ubiquitination plays an important role in regulating endocytosis. Membrane receptor ubiquitination can induce receptor uptake via recruitment of endocytic adaptors, and the adaptors themselves (e.g., Eps15) can be regulated by cycles of ubiquitination and deubiquitination[53]. BTB domain-containing proteins recruit Cullin 3 of the Cullin-RING E3 ubiquitin ligases complexes and are implicated in this process, and many of these BTB proteins contain kelch domains for substrate recruitment (e.g., KLHL8, Fig. 7A)[54]. Apicomplexan K13 contains both BTB and kelch domains, and UBP1 is related to the ubiquitin carboxy terminal hydrolase that targets human Eps15[55], suggesting that both K13 and UBP1 might contribute similar ubiquitin-dependent endocytic function in apicomplexans. The presence of dynamin related protein DrpC at the *Toxoplasma* endocytic site is further suggestive of the membrane fission events of endocytosis, and we observed *Tg*DrpC forming either a ring or constricted puncta just below the K13 ring. *Plasmodium* and other apicomplexans share a DrpC orthologue (Fig. 7D), although it was not identified in the *Plasmodium* cytostome interactome[25]. *Tg*DrpC mutants have previously been implicated with cell replication defects including aberrant mitochondrial morphologies[35], and we cannot exclude possible dual *Tg*DrpC functions including in mitochondrial division. We note, however, that reporter-fused *Tg*DrpC location was sensitive to the terminus being labelled (C-terminal fusions were exclusive to K13 locations, N-terminal fusions include additional apparent distribution including mitochondrial proximity). Furthermore, K13 depletion also resulted in aberrant mitochondrial forms extending beyond the cell so these could represent secondary phenotypes (Fig. 6D).

Other proteins of the apicomplexan endocytic apparatus distinguish it from canonical systems (Fig. 7D). CGAR contains a C-terminal GAR domain that is predicted to bind microtubules and, therefore, might mediate positioning of the endocytic machinery amongst the subpellicular microtubules of apicomplexans. The cysteine protease metacaspase MCA3 is also common to *Plasmodium* (orthologous to *Pf*MCA2) and *Toxoplasma* machineries, although its function here is unclear. The apparent non-involvement of clathrin in endocytosis is also common to the apicomplexans. This classic vesicle coat protein that typically interacts with AP-2 and Eps15 in endocytosis was not detected in reciprocal BioID and pulldown experiments in *Plasmodium*[25,26], as was also the case for our *Toxoplasma* BioID data. Thus, while the involvement of clathrin with AP-1 mediated post-Golgi trafficking has been retained in apicomplexans, endocytosis is clathrin-independent[25,26,32]. In addition to these common derived features of endocytosis, five proteins of the *Plasmodium* cytostome (KIC1, 5, 6, 8 and 9) lacked detectable orthologues in *Toxoplasma*, and ISAP1 is only identifiable in *Toxoplasma* (and other coccidia). Further, while KIC3 does have an orthologue in *Toxoplasma*, we observed this protein in proximity with the basal complex (Fig. S4B) suggesting a different functional role in *Toxoplasma*. So, there is evidence of further specialization of the endocytic structures between these parasites.

The most unusual feature of the endocytic structures of apicomplexans is their permanence. In most cells endocytic events are transient, and signal-activated recruitment to the plasma membrane of the proteins that drive endocytosis is integral to its regulation and control[44–46]. However, we see no evidence of cycles of recruitment and dissociation of proteins to the endocytic structures of *Toxoplasma*. Moreover, the endocytic apparatus is assembled to a seemingly very complete state early in daughter cell formation. This occurs well before these nascent daughter structures are in contact with the plasma membrane that only envelops the daughter cells late in cell division[56]. The inner membrane complex of apicomplexans is an

elaborate structure and its filamentous proteinaceous network that supports the cell, once formed, may not allow the subsequent access and insertion of the endocytic machinery. It is curious, however, that even the dynamin related protein that likely mediates the final steps of membrane fission is present at these very early stages of cell formation. This indicates that DrpC's recruitment is independent of the membrane upon which it will ultimately act. The de novo formation of endocytic structures in daughters means they occur concurrently with those of the mother cell which do not appear to recycle to their daughters. Rather, we see evidence of accumulation of the mother cell structures at the residual body at the base of the interconnected parasites (Fig. 4A). It is of note that several of our BioID false positives were for proteins either near the basal complex or centrosomes (Fig. S4). This might reflect developmental proximity of these structures either during new cell formation when the new IMC pellicle coalesces at the duplicated centrosomes, or during relocation of mother micropores to the cell posterior[40]. K13 in *Plasmodium* is also seen associated with the new cells of the dividing schizont stage, so this preformation of stable endocytic structures during cell division might be a common feature of apicomplexans[24]. While we anticipate that endocytosis is likely to be tightly controlled in apicomplexans as for other eukaryotes, the responses of its molecular components to signalling are seemingly quite different to other models.

The micropore ultrastructure has been observed in apicomplexans by TEM for over half a century[27,56,57] but it has previously not been definitively linked to a cell process. Throughout Apicomplexa the micropore presents as small invaginations of the plasma membrane that are coated by a thin electron dense layer and that penetrates a dense collar that sits within a break in the IMC. Our study defines a protein composition and cellular function to this enigmatic structure at last. The rings formed by K13, Eps15L, MCA3, CGAR and ISAP1 within the sutured boundaries of the IMC alveolae are consistent with the dense collar, and the funneled arrangement of AP-2 components as well as surface protein SAG1 in its lumen are consistent with these proteins lining either side of the membrane invagination. The micropore is observed throughout Apicomplexa as well as in the related myzozoans—dinoflagellates, perkinsids and chrompodellids—and all share the common IMC-type pellicle beneath their plasma membrane[28,29,58–60]. K13 is present in dinoflagellates (and perkinsids and chrompodelids) and the AP-2α/KAE bifurcation is in all alveolates (Figs. 7D and S10). The ring-forming K13, Eps15L, MCA3, ISAP1 and CGAR all contain coiled-coil domains or cytoskeletal interaction motifs, that likely facilitate protein-protein integrations that might contribute to the stability of this structure, and KAE also contains coiled-coil motifs.

Collectively, these data suggest that the micropore is a common endocytic apparatus that evolved in response to the formation of the alveolate cell pellicle. Apicomplexans have seemingly streamlined this apparatus with the loss of the duplicates of K13 and KAE, although the significance of this is unknown. Ciliates have independently evolved an elaborate phagocytic oral apparatus for prey capture that can be orders of magnitudes larger than the micropore. While any evolutionary link to the micropore for this derived structure is unknown, this ciliate-specific adaptation might explain the apparent absence of K13 in ciliates. The *Plasmodium* cytostome, on the other hand, is likely a genuine manifestation of the micropore as strongly suggested by its protein conservation with that of *Toxoplasma*. During the haemoglobin feeding blood-stage, *Plasmodium* disassembles the IMC alleviating it of these obstacles to endocytosis[61]. However, the IMC is reassembled during schizogony and merozoite formation where K13 rings are also seen, and micropores are observed in merozoites by TEM also[24,62,63]. It is likely that these micropores differentiate into cytostomes upon the invasion of new blood cells and ring-stage formation.

The functional significance of endocytosis likely varies in different apicomplexans and different cell stages. In *Plasmodium*, disruption of K13 and other cytostome proteins, either through natural ART-selected mutations or experimental perturbations, results in decreased haemoglobin uptake and a delay to blood-stage parasite growth[24,25]. However, no similar immediate growth delay was seen in *Toxoplasma* with K13-depletion. Endocytosis was not completely ablated in these cells according to our SAG1 assay, and this might reflect either some residual K13 protein, or partial endocytic function even in its absence. But in either case, the dominant phenotype of suppressed endocytosis was a chaotic development of these parasites, an expansion of the plasma membrane such that it and its cytosolic contents spilled in between cells, and the loss of the parasite's ability to egress. Egress of *Toxoplasma* tachyzoites requires completion of cytokinesis by fission of the individual cells' connections to the residual body and the activation of the gliding motility machinery that spans the plasma membrane and IMC. It is likely that both processes are encumbered in these poorly membrane-delineated cells. Our data, therefore, point to plasma membrane homoeostasis, rather than nutrition, as being the more significant function of endocytosis in *Toxoplasma* tachyzoites. Our endocytosis assay shows that surface membrane recycling occurs in the intracellular stages just as it does in the motile extracellular forms[31]. A deficiency of membrane uptake compared to delivery that occurs with secretion events such as surface molecule delivery and dense granule secretion would result in the membrane excess that we observe. Membrane recycling might also be the major role for the micropore of *Plasmodium* merozoites. This transmissive stage between blood cells is not otherwise expected to feed but they do remodel their surface molecules during egress and invasion events. Thus, K13 may contribute to multiple endocytic modalities in *Plasmodium*. Similarly, we cannot eliminate a possible nutritional function for endocytosis in *Toxoplasma*. The observed lower activity of endocytosis of host cytoplasm compared to parasite surface molecules might indicate that it fulfils an alternative role to general nutrition, or its nutritional contribution might be below the threshold of detection in our growth assays. Nevertheless, our identification of a broad range of the molecules of endocytic machineries in apicomplexans, and the development of an effective assay for endocytosis in intracellular stages, provides wide new opportunities to dissect the processes and purposes of this neglected and evidently divergent cell function in these important parasites.

## Methods

### Growth and generation of transgenic *T. gondii*

*T. gondii* tachyzoites from the strain RH and derived strains, including RH Δku80/TATi[64] and RH Δku80/Tir1[65], were maintained at 37 °C with 10% $CO_2$ growing in human foreskin fibroblasts (HFFs) cultured in Dulbecco's Modified Eagle Medium supplemented with 1% heat-inactivated fetal bovine serum, 10 U ml[−1] penicillin, and 10 μg ml[−1] streptomycin, as described elsewhere[66]. When appropriate for selection, chloramphenicol was used at 20 μM and pyrimethamine at 1 μM. Reporter protein-tagging of endogenous gene loci with reporters 3× HA, 3× v5 and eGFP was done according to our previous work[67]. Reporters were C-terminally fused to their proteins of interest unless this was not tolerated (K13 and KAE-fusions could only be recovered when at the N-terminus). For protein function tests by gene knock-downs, endogenous promoters were replaced with an anhydrotetracycline (ATc)-regulatable t7s4 promoter[64] or the proteins where tagged with mini-Auxin-inducible degron (mAID) tag[68] using the same strategy as for the endogenous gene fusions. Oligonucleotides and DNA constructs used for all gene modifications are shown in Supplementary data 3.

### Immunofluorescence microscopy

*T. gondii*-infected HFF monolayers grown on glass coverslips were fixed with 2% formaldehyde at room temperature for 15 min, permeabilized with 0.1% Triton X-100 for 10 min, and blocked with 20%

FBS for 1 h. The coverslips were then incubated with a primary antibody for 1 h, followed by 1 h incubation with a secondary antibody. The primary antibodies used were: anti-HA High Affinity (ROAHAHA, Roche 11867423001); anti-V5 (ThermoFisher R960-25); anti-SAG1 (Thermo-Fisher MA5-18268), anti-GAP45 and anti-IMC1 (both gift from Dominique Soldati-Favre, University of Geneva, Switzerland), all at a dilution 1:250. The secondary antibodies used were: Goat anti-mouse Alexa Fluor 488 (ThermoFisher A-11029), Goat anti-rabbit Alexa Fluor 594 (ThermoFisher A-11012), Goat anti-rat Alexa Fluor 594 (ThermoFisher A-11007), Goat anti-rabbit Alexa Fluor 488 (ThermoFisher A-11008), and Goat anti-rabbit Alexa Fluor 405 (ThermoFisher A-31556), all at a dilution 1:1000. Coverslips were mounted using ProLong Diamond Antifade Mountant (ThermoFisher Scientific, Massachusetts, USA). Images were acquired using a Nikon Eclipse Ti wide-field microscope with a Nikon objective lens (Plan APO, 100×/1.45 oil) and a Hamamatsu C11440, ORCA Flash 4.0 camera. 3D structured illumination microscopy (3D-SIM) was implemented on a DeltaVision OMX V4 Blaze (GE Healthcare, Issaquah, California, USA) with samples prepared as for wide-field immunofluorescence assay (IFA) microscopy with the exception that High Precision coverslips (Marienfeld Superior, No1.5H with a thickness of 170 ± 5 μm) were used in cell culture, and Vectashield (Vector Laboratories, Burlingame, California, USA) was used as mounting reagent. Samples were excited using 405, 488, and 594 nm lasers and imaged with a 60× oil immersion lens (1.42 NA). The structured illumination images were reconstructed in softWoRx software version 6.1.3 (Applied Precision). All fluorescence images were processed using ImageJ version 1.53.

## BioID

**Sample preparation.** For the proximity biotinylation assay, we generated 5 different cell lines (in parental line *T. gondii* tachyzoites RH Δku80) by in situ genomic N- or C-terminal tagging of one of the 5 bait proteins (K13 and KAE tagged at the N-terminus, AP-2α, AP-2μ and DrpC all tagged at their C-terminus) with the promiscuous bacterial biotin ligase, BirA*. The parental cell line was used as a negative control in biotin treatments. We followed the BioID protocol according to Chen et al.[69] and our previous work[19]. Briefly, the parasites were grown in the elevated biotin concentration (150 μM) for 24 prior to egress, separated from the host cell debris and washed 5× in phosphate-buffered saline. The cell pellets were lysed in RIPA buffer by sonication and the lysates containing ~5mg of total protein content were incubated with 250 μl of unsettled Pierce™ Streptavidin Magnetic Beads (Thermo-Fisher: 88817) overnight at 4 °C with gentle agitation. The beads were then sequentially treated as follows: washed 3× in RIPA, 1× in 2 M UREA and 100 mM triethylammonium bicarbonate (TEAB; Sigma); washed 3× in RIPA; reduced in 10 mM DTT and 100 mM TAEB for 30 min at 56 °C; alkylated in 55 mM iodoacetamide 100 mM TAEB for 45 min at room temperature in the dark; and washed in 10 mM DTT 100 mM TAEB, followed by 2× 15 min in 100 mM TAEB with gentle agitation. The peptides were digested on the beads by a 1 h 37 °C incubation in 1 μg of trypsin dissolved in 100 mM TAEB, followed by an overnight 37 °C incubation after adding an extra 1 μg of trypsin.

**TMT labelling.** The peptide concentrations were measured using the Pierce™ Quantitative Fluorometric Peptide Assay (ThermoFisher: 23290) according to the manufacturer's instructions and 5 μg of the digested peptides were subjected to the tandem mass tag (TMT) labelling using TMT10plex isobaric tagging reagent set (ThermoFisher: 90110). Each TMT reagent vial containing 0.8 mg of the labelling reagents was brought to room temperature and dissolved in 60 μl of LCMS-grade acetonitrile immediately before use. The TMT reagents were then split to three sets without exceeding the labelling capacity, and 20 μl of the TMT reagents were added to each peptide sample. After incubating for 1 hour at room temperature, 8 μl of 5% hydroxylamine (v/v) was added to each sample, followed by incubation for

15 min to quench the reaction. The TMT-labelled fractions were combined and dried in a vacuum centrifuge (Labconco) at 4 °C.

**Analysis of TMT-labelled peptides by liquid chromatography and tandem mass spectrometry.** LCMS analyses were carried out on an Orbitrap™ Fusion™ Lumos™ Tribrid™ mass spectrometer coupled on-line with a Dionex Ultimate™ 3000 RSLCnano system (Thermo Fisher Scientific) as previously described[67]. The XCalibur v3.0.63 software was used to control the instrument parameters and operation, and record and manage the raw data. The LCMS system was operated in the positive-ion data-dependent acquisition mode with the SPS-MS[3] acquisition method with a total run time of 120 min. The dried TMT10plex-labelled peptide samples resolubilized in an LC-MS sample loading solution (0.1% aqueous formic acid) at a concentration of approximately 1 μg/μl. Approximately 1 μg of the sample was loaded onto a micro-precolumn (C18 PepMap 100, 300 μm i.d. × 5 mm, 5 μm particle size, 100 Å pore size, Thermo Fisher Scientific) with the sample loading solution for 3 min. Following the loading step, the valve was switched to the inject position, and the peptides were fractionated on an analytical Proxeon EASY-Spray column (PepMap, RSLC C18, 50 cm × 75 μm i.d., 2 μm particle size, 100 Å pore size, Thermo Fisher Scientific) using a linear gradient of 2–40% (vol.) acetonitrile in aqueous 0.1% formic acid applied at a flow rate of 300 nl/min for 95 min, followed by a wash step (70% acetonitrile in 0.1% aqueous formic acid for 5 min) and a re-equilibration step. Peptide ions were analyzed in the Orbitrap at a resolution of 120,000 in an *m/z* range of 380–1500 with a maximum ion injection time of 50 ms and an AGC target of 4E5 (MS[1] scan). Precursor ions with the charge states of 2–7 and the intensity above 5000 were isolated in the quadrupole set to 0.7 *m/z* transmission window and fragmented in the linear ion trap via collision-induced dissociation (CID) at a 35% normalized collision energy, a maximum ion accumulation time of 50 ms and an AGC target of 1E4 (MS[2] scan). The selected and fragmented precursors were dynamically excluded for 70 s. Synchronous precursor selection (SPS) was applied to co-isolate ten MS[2]-fragments in the linear ion trap with an isolation window of 0.7 *m/z* in the range of *m/z* 400–1200, excluding the parent ion and the TMT reporter ion series. The SPS precursors were activated at a normalized collision energy of 65% to induce fragmentation via high-collision energy dissociation (HCD). The product ions were measured in the Orbitrap at a resolution of 50,000 in a detection range of *m/z* 100–500 with a maximum ion injection time of 86 ms and an AGC of 5E4 (MS[3] scan). The mass spectrometry proteomics data have been deposited to the ProteomeXchange Consortium via the PRIDE[70] partner repository with the dataset identifier PXD034193.

**Raw LCMS data processing.** The processing of raw LSMS data for peptide and protein identification and quantification was performed with Proteome Discoverer v2.3 (Thermo Fisher Scientific). Raw mass spectra were filtered converted to peak lists by Proteome Discoverer and submitted to a database search using Mascot v2.6.2 search engine (Matrix Science) against the protein sequences of *Homo sapience* (93,609 entries retrieved from UniProt on 09.04.2018), *Bos taurus* (24,148 entries retrieved from UniProt on 17.04.2017), and *T. gondii* strain ME49 (8,322 entries retrieved from ToxoDB.org release 36 on 19.02.2018)[71]. Common contaminant proteins—e.g., human keratins, bovine serum albumin, porcine trypsin—from the common Repository of Adventitious Proteins (cRAP, 123 entries, adapted from https://www.thegpm.org/crap/) were added to the database, as well as the sequence of the BirA* used to generate the BioID bait proteins by gene fusion. The precursor and fragment mass tolerances were set to 10 ppm and 0.8 Da, respectively. The enzyme was set to trypsin with up to two missed cleavages allowed. Carbamidomethylation of cysteine was set as a static modification. The dynamic modifications were TMT6plex at the peptide N-terminus and side chains of lysine, serine, and threonine, oxidation of methionine, deamidation of asparagine and glutamine,

and biotinylation of the peptide N-terminus or lysine side chain. The false discovery rate of peptide-to-spectrum matches (PSMs) was validated by Percolator v3.02.1[72] and only high-confidence peptides (FDR threshold 1%) of a minimum length of 6 amino acid residues were used for protein identification. Strict parsimony was applied for protein grouping.

TMT reporter ion abundances were obtained in Proteome Discoverer using the most confident centroid method for peak integration with 20 p.p.m. tolerance window. The isotopic impurity correction as per the manufacturer's specification was applied. For protein quantification, PSMs with precursor co-isolation interference above 50% were discarded, and the TMT reporter ion abundances determined for unique (sequence found in proteins belonging to a single protein group) and razor (if sequence is shared by protein belonging to multiple protein groups, the quantification result is attributed to the best-associated Master Protein) peptides were summed.

**Statistical analysis of protein enrichment in BioID.** Data analysis was performed with *R* v3.6[73] using packages *tidyverse* v1.2.1[74] for data import, management, and manipulation, *Bioconductor*[75] packages *MSnbase* v2.10.1[76] for managing quantitative proteomics data, *biobroom* v1.16.0 (https://github.com/StoreyLab/biobroom) for converting *Bioconductor* objects into *tidy data frames*, and *limma* v3.40.6[77] for linear modelling and statistical data evaluation.

The protein-level report generated by Proteome Discoverer was imported into R and filtered to remove non-*Toxoplasma* and low-confidence (protein FDR confidence level "Low", $q \geq 0.05$). Only Master Proteins with a complete set of TMT abundance values across three replicates of the BioID bait and control samples were considered for the analysis. The protein abundance values in each biological sample were corrected for the total amount using normalization factors derived from the abundances of two proteins, acetyl-CoA carboxylase ACC1 (TGME49_221320) and pyruvate carboxylase (TGME49_284190). Both proteins are highly expressed, endogenously biotinylated, and reside in the matrix of subcellular compartments, the apicoplast and mitochondrion for ACC1 and PC, respectively, where they are not accessible to the BirA*-fused BioID baits. Hence, these two proteins could serve as suitable internal standards. The normalized protein abundances were log2-transformed and modelled as a simple linear relationship between the abundances described by a constant factor (intercept), which is of no interest to us, and the condition (BirA*-tagged vs. control) using limma. If the condition parameter estimated by limma linear model was significantly different from zero we concluded that the condition (presence of the BirA*-fused bait) had a significant effect on the protein abundance. Also, limma estimated the model parameters taking into account the relationship between protein average intensities and the variance (low-abundance proteins tend to have a greater variance) by empirical Bayesian shrinking of the standard errors towards the trend. This enabled a better control of false discoveries and outliers affording more robust identification of significantly enriched proteins. The resulting *p*-values were adjusted for multiple testing using the Benjamini-Hochberg method (FDR < 1%). Proteins with the adjusted *p*-value below 0.01 were deemed significantly changing abundance in the BioID bait condition vs. the control, and those of them whose abundance in the BioID bait condition was more than two-fold greater than in the control were considered significantly enriched.

**Plaque assay**
To test lytic cycle competence of knockdown cell lines by plaque formation in HFF monolayers, 500 freshly lysed parasites were added to T25 flasks containing HFF monolayer. 0.5 μg/ml of ATc was added to induce the gene knockdown, or omitted for uninduced controls. After 7 days of growth, flasks were aspirated, washed once with PBS, fixed with 5 ml of 100% methanol for 5 min and stained with 5 ml of crystal

violet solution for 15 min. After staining, the crystal violet solution was removed, and the flasks were washed three times with PBS, dried and imaged. All protein depletion plaque assays were performed three times independently.

**Replication assay**
For replication assays, the parasites were pre-treated with ATc for 48 h before the egress from the host cell and subsequent invasion of the HFF monolayer growing on coverslips in 6-well plates. Parasites were allowed to grow for further 24 h with ATc. Parasitophorous vacuoles were scored containing either 1, 2, 4, 8 or 16 parasites. A minimum of 200 parasitophorous vacuoles were scored for each of the three biological replicates. *P* values were calculated with the Student's *T*-test using STATA v.14, and the bar graphs were drawn using GraphPad Prism, v8 (GraphPad, California USA).

**Host cytosolic protein uptake assay**
Inducible mCherry HeLa cells previously described[48] were seeded into 6-well plates at a density of $1.5 \times 10^5$ cells per well. Cytosolic mCherry expression was induced for 4 days by adding 2 μg mL$^{-1}$ of doxycycline each day. Prior to infection of host cells expressing cytosolic reporter, parasites were treated with 0.5 μg mL$^{-1}$ anhydrotetracycline (ATc) or vehicle control for 48 h. Additionally, parasites were treated with 5 μM of the protease inhibitor LHVS for 24 h to inhibit the degradation of material delivered to the VAC. Cells were infected with $1.0 \times 10^6$ parasites and allowed to replicate for 24 h in the presence of ATc or vehicle control and LHVS. Parasites were harvested for analysis by scraping and syringing the infected monolayer on ice followed by filtration and centrifugation at $1500 \times g$ for 10 min at 4 °C. Isolated parasites were then treated with a 1 mg mL$^{-1}$ pronase and 0.01% saponin-1× PBS solution for 1 h at 12 °C, centrifuged and washed 3× before adding to Cell-Tak coated slides. Extracellular parasites were fixed with 4% paraformaldehyde for 10 min and permeabilized with 0.1% Triton X-100 for 10 min prior to imaging. At least 200 parasites were analyzed per sample and the percentage of parasites positive for mCherry was quantified by dividing the number of parasites containing mCherry signal derived from the host cytosol divided by the total number of parasites analyzed.

**Uptake of parasite surface protein assay**
**Internal tagging of SAG1 (SRS29B TGME49_233460) by transient CRISPR-Cas9 expression.** SAG1-Halo and iΔHA-*Tg*K13-SAG1-Halo were generated by tagging SAG1 in the Δku80 and Δku80 iΔHA-*Tg*K13 line using transient Cas9 transfection targeting the following gRNA: TGCAGCCCCGGCAAACTCCAC(GGG). The strain was generated as previously described for Cas9 tagging[78]. Briefly, gRNA oligos were annealed and ligated into the Cas9 vector and verified by sequencing (Eurofins Genomics). Reparation template DNA were generated by amplifying the halo with homology arm (50bp) with SAG1 by PCR using Q5 High-Fidelity DNA Polymerase (New England BioLabs). The repair template was purified using a PCR purification kit (Blirt). Parasite transfection, sorting and screening for positive mutants was done as previously[78]. Briefly, newly released RHDiCreΔku80 or Tet-K13-Ha Δku80 tachyzoites were transfected with the repair template and 10–12 μg of SAG1-Cas9-YFP. The parasites were mechanically egressed 24 to 48 h after transfection, passed through a 3 μm filter, and those transiently expressing Cas9-YFP enriched via FACS 23 (FACSARIA III, BD Biosciences) into 96-well plates (a minimum of 3 events per well). Resultant clonal lines were screened by IFA for SAG1 labelling and integration was confirmed by PCR.

**Specific labelling of the plasma membrane SAG1.** SAG1-Halo strain was labelled with either the non-permeable Alexa 660 (1/1000) or permeable Oregon green (1/4000) dyes (Promega) in cold media for 1h. Parasites were then washed prior to transfer to Ibidi live-cell dishes

(29 mm) coated with 0.1% poly-L-Lysine as previously described[31]. After letting the parasite settle for 15 min, parasites where imaged live on a Leica-DMi8 wide-field microscope attached to a Leica DFC9000 GTC camera, using a 100× objective to compare the difference between permeable and non-permeable dye labelling.

**SAG1-endocytosis assays.** For extracellular tachyzoites, the SAG1-Halo strain was labelled with the non-permeable Alexa 660 dye (1/1000) in cold media for 1 h. Parasites were wash 3× to remove excess ligand. Parasites were incubated at 4 °C or 37 °C on the FBS coated live-cell dishes (Ibidi live-cell dishes, 29 mm) for 1 h prior to live imaging, as described below, for evaluation of internalization of the SAG1 signal. For SAG1 recycling assays of intracellular tachyzoites, parasites were labelled with Alexa 660 as above and then inoculated onto HFF monolayers in Ibidi dishes. Parasites were allowed to replicate for 24 h prior to imaging. For endocytosis assays with K13-depletion, iΔHA-*Tg*K13-SAG1-Halo parasites were first induced for 48 h with or without ATC, before mechanical egress, filtering and labelling. Parasites were then allowed to reinfect new HFFs cells, grow for 24 h under the same ATc treatment and were then imaged live, as described below, for evaluation of internalization of the SAG1 signal. Endocytic activity was assessed by the presence of the PM-SAG1 vesicles (non-permeable halo-Alexa Fluor®-660) and the percentage of parasites showing the presence of vesicles was determined. Mean values of three independent experiments ± SD were determined. The mean number of PM-SAG1 vesicles/parasite was determined from the total number of vesicles inside vacuoles divided by the number of parasites in the vacuole. For membrane accumulation, we scored aggregation of PM-SAG1 as signal in the residual body or outside the typical parasite plasma membrane. For these analyses at least 25 vacuoles per replicates were used and mean values of three independent experiments ± SD were determined. All SAG1-Halo images were acquired on a Leica-DMI8, objective 100x with the LasX software (v3.7.4). Imaged were deconvolved using Huygens essential software (v18.04) and batch express processing. Fiji (v1.53c) was used to analyze the picture and all count were made manually.

**Egress assay**

iΔHA-*Tg*K13 HP03-eGFP parasites were seeded in a T25 flask and treated with ATc for 48 h or vehicle (DMSO) for the negative control. After 48 h the extracellular parasites were removed, and the intracellular parasites were mechanically egressed by needle-passed, reseeded into the ibidi live imaging chamber, and then grown for a further 24 h with ATc or DMSO. Images of 10 fields of view were taken using the transmembrane protein HP03-eGFP as a marker for the parasites. Next, 5 μM of the Ca²⁺ ionophore A23187 was added and incubated for 10 min at 37 °C before the same 10 fields of view were imaged. The number of vacuoles and parasites per vacuole were then counted in the first set of images (before adding the Ca²⁺ionophore) and the percentage of successfully egressed parasites quantified. *P* values were calculated with the Student *T*-test using STATA v.14, and the bar graphs were drawn using GraphPad Prism, v8 (GraphPad, California, USA).

**Electron microscopy**

**Immuno-TEM.** 3v5-mAID-K13 egressed parasites were harvested by centrifugation, washed with phosphate buffer (0.2 M Na₂HPO₄, and 0.2 M NaH₂PO₄ with a Ph of 7.24*)*, and then fixed for 30 min at RT (4% formaldehyde). The subsequent processing was performed at 4 °C where cells were dehydrated in 30%, 50% and 70% ethanol (2× for 5 min each), infiltrated with hard LRW resin (Agar Scientific) that was cured in an embedding oven at -55 °C overnight. Ultrathin sections were cut using a Leica Ultracut ultramicrotome and collected on 200 mesh nickel grids covered with a carbon film (EM resolutions). The grids were blocked for 30 min at RT in 0.8% BSA/PBS/0.01% Tween 20 (blocking buffer) then incubated with the primary antibody (anti-V5

mouse, ThermoFisher R960-25) diluted in blocking buffer at 1:10 in a humid chamber at 4 °C overnight. The grids were washed 3× for 5 min each in washing buffer (PBS/0.01% Tween 20) and then incubated with the goat anti-mouse IgG H&L (10 nm Gold) secondary antibody (Abcam ab39619, 1:20 in blocking buffer) for 1h at RT. Negative controls were incubated with the secondary antibody only. Grids were washed 3x for 5 min each in a drop of washing buffer, then passed over two drops of PBS and fixed on a drop of 3% glutaraldehyde/PBS for 10 min. Then the grids were washed 3× in deionized water, allowed to air dry and were then post-stained in 2% aqueous uranyl acetate and Reynold's lead citrate for 5 min each. Samples were viewed in a Tecnai G2 transmission electron microscope run at 200 keV with a 20 μm objective aperture for improved contrast and images were acquired using an AMT CCD camera.

**Ultrastructural TEM.** For iΔHA-*Tg*K13-SAG1-Halo parasites with or without K13 depletion, the parasites were induced for 48 h with or without ATc, released mechanically and filtered prior transfer to Ibidi μ-dishes previously seeded with HFF cells. After 24 h of replication the parasites were fixed with 2.5% glutaraldehyde in 0.1 M phosphate buffer pH 7.4. The parasites were washed three times at room temperature with PBS (137 mM NaCl₂, 2.7 mM KCl, 10 mM Na₂HPO₄, 1.8 mM KH₂PO₄, pH 7.4) and post-fixed with 1% (w/v) osmium tetroxide for 1 h. Subsequent to washing with PBS and water, the samples were stained en bloc with 1% (w/v) uranyl acetate in 20% (v/v) acetone for 30 min. Samples were dehydrated in a series of graded acetone and embedded in Epon 812 resin. Ultrathin sections (thickness, 60 nm) were cut using a diamond knife on a Reichert Ultracut-E ultramicrotome. Sections were mounted on collodium-coated copper grids, post-stained with lead citrate (80 mM, pH 13) and examined with an EM 912 transmission electron microscope (Zeiss, Oberkochen, Germany) equipped with an integrated OMEGA energy filter operated in the zero-loss mode at 80 kV. Images were acquired using a 2k × 2k slow-scan CCD camera (Tröndle Restlichtverstärkersysteme, Moorenweis, Germany).

**Ultrastructural SEM.** DrpC-mAID-3v5 and wildtype parasites were grown with auxin treatment (500 μM) for 48 h, mechanically egressed and then reinfected fresh host cells in 35 mm ø Ibidi dishes with plastic coverslips. A further 24 h of growth was allowed with continued auxin treatment and then cells were processed for EM in these culture dishes. Cells were fixed in 2% glutaraldehyde/2% formaldehyde in 0.05 M sodium cacodylate buffer pH 7.4 containing 2 mM calcium chloride overnight at 4 °C. After washing 5× with 0.05 M sodium cacodylate buffer pH 7.4, samples were osmicated (1% osmium tetroxide, 1.5% potassium ferricyanide, 0.05 M sodium cacodylate buffer pH 7.4) for 3 days at 4 °C, washed 5x in DIW (deionised water), then treated with 0.1% (w/v) thiocarbohydrazide/DIW for 20 min at room temperature in the dark. After washing 5× in DIW, samples were osmicated a second time for 1 hour at RT (2% osmium tetroxide/DIW), washed 5× in DIW, then blockstained with uranyl acetate (2% uranyl acetate in 0.05 M maleate buffer pH 5.5) for 3 days at 4 °C. Samples were washed 5× in DIW and then dehydrated in a graded series of ethanol (50%/70%/95%/100%/100% dry) and 100% dry acetonitrile, 3× in each for at least 5 min. Samples were infiltrated with a 50/50 mixture of 100% dry acetonitrile/Quetol resin (without Benzyldimethylamine [BDMA]) overnight, followed by 3 days in 100% Quetol (without BDMA). Then, the sample was infiltrated for 5 days in 100% Quetol resin with BDMA, exchanging the resin each day. The Quetol resin mixture is 12 g Quetol 651, 15.7 g NSA, 5.7 g MNA and 0.5 g BDMA (all from TAAB). The Ibidi dishes were filled with resin to the rim, covered with a sheet of Aclar and cured at 60 °C for 3 days. The sample blocks were then cut from with a hacksaw and ultrathin sections cut and collected on 300 mesh copper grids. Samples were imaged in a Verios 460 scanning electron microscope (FEI/Thermo Fisher Scientific) run at an accelerating voltage of 4 keV

and 0.2 nA probe current using the concentric backscatter detector in field-free mode (low magnification) or immersion mode (high resolution).

### Binding of α ear appendage domains to WxxF motif of *Tg*Eps15L

The construction of a GST fusion of *Mm*AP-2α ear appendage domain has been previously described[79] and encompasses amino acids 695–938 of mouse AP2A2. Equivalent domain boundaries for *Tg*KAE (TGME49_272600) and *Tg*AP-2α (TGME49_221940) ear appendage domains were selected based on homology to MmAP-2α. *Tg*KAE (amino acids 1406–1672 end) and TgAP-2α (amino acids 1015–1348 end) appendage domains were made synthetically and codon optimized for expression in *Escherichia coli*; adding BamH1 and Xho1 sites for *Tg*AP-2α and BamH1 and Sal1 for *Tg*KAE for in-frame cloning into pGEX4T-1. For fusion protein production, the DNA was transformed into BL21(DE3) competent cells for high protein expression and induced mid-log phase with 0.4 mM IPTG 16–20 h at 22 °C. Fusion proteins were recovered using Glutathione-Sepharose 4B (GE Healthcare) and eluted from the beads using 30 mM reduced glutathione. His-tagged fragment of *Tg*Eps15L (TGME49_227800) encompassing amino acids 897–1023 (952-WxxF), whose boundaries were predicted using DomPRED (URL:http://bioinf.cs.ucl.ac.uk/software.html) was made synthetically and codon optimized for expression in *E. coli*. The fragment was cloned using BamH1 and EcoR1 (adding a stop codon) into pTrcHisA. For fusion protein production, the DNA was transformed into BL21(DE3) competent cells (Invitrogen), induced mid-log phase with 0.4 mM IPTG, purified using Ni-nitrilotriacetic acid agarose beads (QIAGEN) and eluted with 250 mM imidazole. Point mutation W952A (952-AxxF) was made using site-directed mutagenesis.

For the far-western blot assay, 2.5 µg of the His-tagged fusion proteins were subjected to SDS-PAGE on 16% Bolt Bis-Tris gel (ThermoFisher) in MOPS running buffer gels and blotted onto a nitrocellulose membrane. The blot was blocked with 20 mM Tris-HCl, pH 7.5, 150 mM NaCl, 0.05% Tween 20, 0.5% BSA, 3 µM reduced glutathione for 30 min and this buffer was used in all the following steps. The blot was incubated with 10 nM GST, GST-MmAP-2α, GST-TgAPale or GST-TgAP-2α for 1h, washed for 30 min, and then labelled with 1:30,000 anti-GST (Invitrogen 700775) followed by 1:10,000 anti-rabbit-HRP (Invitrogen 31460) and developed using ECL Prime (GE Healthcare) and X-ray film.

### Structural predictions

The 3D structure of *Tg*KAE ear appendage domain was modelled using SWISS-MODEL[80] by using as template the mouse Alpha-adaptin Appendage Domain (PDB 1w80; chain A, sequence identity 31%). Using the positions of the bound peptides (FxDxF and WxxF) in 1w80, a resultant *Tg*KAE peptide complex structure was rebuilt and further energy minimized in coot[81]. Figures for the protein structures were drawn with Pymol[82], and the protein-protein interaction networks were generated with Ligplot+[83].

### Homology searching, domain detection and phylogenetic analyses

Protein homologues were searched with the iterative version of the profile hidden Markov models (HMMs) search engine (jackhammer)[84]. Conserved domains were detected with InterProScan[85] and the similarity levels between the *T. gondii* and human BTB, Kelch, Adaptin_N and α_C2/αC (ear) domains were evaluated using a pairwise protein BLAST (https://blast.ncbi.nlm.nih.gov/Blast.cgi). For the phylogenetic analyses, sequences were aligned using Mafft v7.407 with the L-INS-i algorithm[86]. Phylogenetic analyses were performed locally and using the CIPRES online portal[87]. Alignments were masked and trimmed manually using Mesquite v3.6 (https://www.mesquiteproject.org) or Jalview (https://www.jalview.org). Bayesian analysis was carried out using MrBayes v3.2.7a35 hosted on CIPRES. Datasets were run under a mixed model with four independent runs of four chains each, sampling

every 500 generations up to a total of either 1,000,000 or 10,000,000 MCMC generations, depending on convergence criteria. Trees were summarized discarding the first 20% of samples as burn-in; convergence was assessed by manually expecting the generation vs. log-likelihood plots for stationarity, as well checking parameter PSRF values for various parameters. Maximum-likelihood (ML) rapid boostrapping was carried out using RAxML v8.2.1232 (LG+Γ model, rapid bootstrapping) hosted on CIPRES or locally using PhyML-3.1 or IQ-TREE v1.6.12 (1000 bootstraps, with the optimal substitution model)[88,89]. SH-like aLRT and aBayes branch support values were obtained in PhyML-3.1.

### Statistics and reproducibility

All experiments were performed as three independent replicates. The details on statistical analyses used are provided in figure legends where applicable. To identify the differentially enriched proteins in the BioID experiments (Supplementary data 2), the normalized log2-transformed protein abundance values measured for the BirA*-tagged and negative control samples were compared using a moderated one-sided *t*-test in the limma linear modelling framework. To compute the moderated t-statistics, the variance estimates were moderated across proteins using an empirical Bayes method of shrinking the protein-wise residual variances towards a global trend. The computed p-values were adjusted for multiple comparisons according to the Benjamini-Hochberg method. The results of the experiments that were not statistically analyzed (immunofluorescence assays, electron microscopy analyses, western blot analyses; Figs. 1, 2B, 3A, B, 4A, B, 6D, 7C, Figs. S1, S2A,C-E, S3, S4B, S5A, B, S6A, B, S7A, S9C) were similar for all replicates and representative images are shown for each experiment. For microscopy analyses, over 100 individual parasites/vacuoles were observed for each slide (i.e. a single replicate), and the results were consistent for each replicate.

### Reporting summary

Further information on research design is available in the Nature Portfolio Reporting Summary linked to this article.

## Data availability

The mass spectrometry proteomics data have been deposited to the ProteomeXchange Consortium via the PRIDE partner repository with the dataset identifier PXD034193. ToxoDB accession numbers of the BioID hits and the localized proteins are shown directly in the corresponding figures (Figs. 2, and S4). The sequence identifiers for the protein sequences that were used for phylogenetic analyses are shown for each taxon in the phylogenetic trees (Figs. S10 and S11). The structure of the mouse Alpha-adaptin Appendage Domain (identifier 1w80) was retrieved from the Protein Data Bank (PDB). Source data are provided with this paper.

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

## Acknowledgements

This work was supported by the Wellcome Trust, United Kingdom, Investigator award 214298/Z/18/Z to R.F.W., a Deutsche For-schungsgemeinschaft (DFG, German Research Foundation) grant 464878930 to S.G., and a DFG Equipment grant INST 86/1831-1 to Prof. Markus Meissner. B.N.M.S. was supported by a Gates Cambridge Scholarship, and K.B. was supported by the Leverhulme Early Career Fellowship (ECF-2015-562) provided jointly by the Isaac Newton Trust and the Leverhulme Trust. M.S.R. and J.H. was supported by Wellcome Trust grants 086598 and 214272 to M.S.R. and a Wellcome Trust Strategic Award 100140 to the Cambridge Institute for Medical Research (CIMR). N.R.Z. was supported by Wellcome Trust grant WT 207455/Z/17/Z. C.M.K. was supported by a Canada Vanier Graduate Scholarship and J.B.D. is supported by the Canada Research Chair (Tier II) in Evolutionary Cell Biology and by the Natural Sciences and Engineering Research Council of Canada (RES0043758, RES0046091). Y.R-C. was supported by NIH awards F31AI152297 and T32AI007528. We thank VEuPathDB for their invaluable Informatics Resources, and Markus Meissner, Christine Hopp, Nicola Hodson, Julian Rayner, Mirko Singer and David Warhurst for useful discussions, and Cornelia Niemann and Karin Müller for assistance with electron microscopy.

## Author contributions

Conceptualization, R.F.W, L.K., and S.G.; Methodology, L.K., B.N.M-S, K.B., and S.G.; Formal analysis, K.B. and N.R.Z.; Investigation, L.K., B.N.M-S., C.M.K., S.B., J.H., Y. R-C., V.J.C.H., A.K. and S.G.; Writing – original draft, R.F.W and L.K.; Writing – review & editing, L.K., B.N.M-S, K.B., S.B., J.H., Y.R-C., N.R.Z., J.B.D., V.B.C., M.S.R., S.G. and R.F.W; Visualization, L.K., B.N.M-S., C.M.K, K.B., J.H., Y.R-C., N.R.Z. and S.G.; Supervision, R.F.W., S.G., M.S.R., V.B.C. and J.B.D.; Funding acquisition, R.F.W., S.G., M.S.R., V.B.C., and J.B.D.

## Competing interests

The authors declare no competing interests.
