## [Peer Review File · Nature Communications]

Stable endocytic structures navigate the complex pellicle of apicomplexan parasitesReviewers' Comments:

Reviewer #1:

Remarks to the Author:

Koreny et al

NCOMMS-22-26335-T

This manuscript explores the relationship of the Kelch-13 centered complex and the IMC and pellicle of *T. gondii* that form fixed sites of endocytosis that are situated in gaps of the inner membrane complex. The core of the complex consists of Kelch 13 and AP2 adapter proteins that are associated with clathrin mediated endocytosis in other eukaryotes. Kelch13 and AP2 proteins have been previously characterized as components of the cytosome of *Plasmodium falciparum*, which is used by malaria parasites for hemoglobin uptake. Kelch-13, a core component of the complex, and other cytosome-associated proteins are implicated in artemisinin resistance.

The work uses multiple complementary techniques to understand the structure and organization of this complex, including 3D SIM super resolution microscopy and proteomics. The images are high quality and the overall model is convincing, but the manuscript suffers throughout because of qualitative rather than quantitative support for the observations.

The identification of a novel endocytic structure that is fixed and composed of both conserved and novel proteins is significant, particularly since the proteins appear to be conserved in Apicomplexa and essential for parasite survival.

1. The abstract could be rewritten to focus on the novel cell biology features of the findings. Yes, the endocytic complex proteins are implicated in artemisinin resistance in malaria, but this isn't particularly relevant for the current story that focuses upon *Toxoplasma* and identifies a conserved endocytic complex in Apicomplexa that is fixed and localized to the inner membrane complex.
2. Figure 1 and S1 show localization data of K13, 5 members of the AP2 associated family, and DrpC, a member of the dynamin family. The AP2 localize with K13 though Beta also also has other localization. They report DrpC is seen in 2 forms one similar to K13 and the other a conical punctum of smaller diameter consistent with constriction. How many images, replicates were used to define these localizations? Did they perform quantitation of localizations, quantify distances or perform statistical analyses?
3. Figure 2 shows localization of proteins identified by BioID with 5 bait proteins. IMC proteins were amongst the interacting proteins identified. Details of the criteria to fold change are presented as a heat map, but the details of the actual data are not provided, and the heat maps are very difficult to read with tiny fonts. From the methods provided, the analysis appears appropriate. The authors state the data have been deposited, but as presented in the manuscript the rigor and validity of the analysis are impossible to assess because of incomplete information. Further details of false and true positive data should be provided in a supplementary table.
4. Figure 3 explores the positioning of the complex in relationship to the IMC. K13 localizes to longitudinal or transverse plate boundaries, and regions where IMC membranous markers are excluded, consistent with coordinated interruption of the IMC, with K13 providing an interface between plasma membrane and cytosol. Again, details of replicates, images analyzed and quantitation of these observations is not provided.
5. Figure 4 Live fluorescent imaging reveals stable complexes and that the complexes appear to assemble during daughter cell formation prior to plasma membrane association with the IMC. Again no quantitation of these observations is provided. Plaque assays have no statistics or information about replicates.
6. Figure 5 presents a new endocytosis assay based upon Halo tagging of SAG1 that can be used to quantify external vs internalized SAG1. Using this assay, they show that K13 Knockdown is associated with decreased endocytosis of SAG1. Replicates of assays are not provided nor is any quantitation or statistics.
7. Figure 6 shows that K13 KD doesn't affect parasite replication, but vacuoles are disorganized with impaired plasma membrane marker localization. Does K13 KD affect egress or subsequent invasion?

Reviewer #2:

Remarks to the Author:

This study is centered around the TgK13 complex and its function in parasite endocytosis. The authors first use proximity labeling and high-resolution microscopy to identify and localize several components of the parasites endocytic machinery that likely corresponds to the micropore. They then develop a novel endocytosis assay based around the recycling of SAG1 and show that K13 is important for this dynamic process. Finally, the authors characterize the defects of K13-depleted parasites to demonstrate that the purpose of endocytosis in these parasites is to maintain plasma membrane homeostasis rather than uptake nutrition for growth. Overall, this is an important study that sheds light into a long-standing mystery with extensive techniques and robust evidence to support the conclusions. Specific comments are below.

Major comments

1. Overall, this is solid and important work. The identification of a group of proteins that are almost certainly the micropore is nicely done with excellent imaging and the use of proximity labelling with multiple bait proteins. In addition, the importance of the development of the endocytosis assay that assesses the recycling of SAG1 is important. Existing assays are tricky and hard to quantify and this advance provides a new method for assessing uptake via these proteins.
2. The authors never directly show that the protein machinery identified corresponds to the micropore. Could one of these proteins be directly localized to the micropore to firmly resolve this? (the reviewer acknowledges this may be technically challenging).
3. The authors mention that many of these components contain cytoskeletal or protein-protein interaction domains that might make the micropore a more stable structure. Is K13 tethered to the cytoskeleton as assessed by detergent fractionation (as done for ISAP1 and its interactors in Chern et al.)
4. The authors should better acknowledge the findings of previous work, in particular the overlap with Chern et al. which localized several of the same proteins and also showed localization to the IMC suture junctions (note that this does not diminish the findings of this paper which provides additional proteins and a much higher resolution of the components within the micropore and surrounding components of the IMC).
 - Chern et al. previously reported the localization of ISAP1 and its partners - KAE (TGME49_272600), EPS15L (TGME49_227800), and CGAR (TGME49_297520) and showed colocalization to the sutures.
 - DrpC was previously localized to discrete puncta in three separate papers (Melatti et al. 2019, Heredero-Bermejo et al. 2019, Chern et al. 2021)
 - These findings should be included in the results (lines 168, 181-185), added to the previously localized proteins in Fig 2A, and noted in Fig 3 (lines 227-237).
5. Regarding epitope tagging endocytic machinery components
 - a) Fig 1 - The authors should state which tags were used for each protein and whether they were N or C terminally tagged in the figure or in the figure legend. Same applies for other figures when needed or if changed for a particular experiment. This could also be accomplished with a supplementary table.
 - b) It appears that N-terminal tagging was used in some cases (for K13 and KAE BioID at least). Is this because C-terminal tagging gave apparent mislocalization or inactivity of the fusion protein? – or was some feature of the protein that was avoided? If so, this should be discussed.

6. How does K13 depletion affect the localization of its partners? Are they still present in foci at the IMC sutures? or is the complex disrupted and the proteins are then mislocalized?

7. Since the authors already have knockdown-inducible lines for other proteins, is it possible to show that KD of other components (or even a single other one) show similar plasma membrane homeostasis problems? Additional functional data would provide more compelling evidence of this paradigm shift for the function of the micropore in apicomplexans.

8. Fig 6 "K13-depletion disrupts parasite order and integrity rather than replication rate" The authors state that replication rate is not affected - but then presumably shortly after the timepoint examined, the parasites die as no plaques are formed. Is this likely due to an incomplete knockdown at this stage? Discuss how protein levels and the phenotype ultimately affect viability?

Minor comments

1. DrpC is not enriched very highly in multiple datasets compared to other components of the K13 complex.

- Melatti et al has shown DrpC associated with the mitochondria and Chern et al sees DrpC signal distinct from ISAP1. Is DrpC always K13 associated? I believe all of these were C-terminal tags thus unlikely to be the N-terminal tagging issue brought up in the paper?

2. Fig 2A, S3. The authors find several centrosome-like proteins and basal complex proteins in their BioID experiments. It would be stronger (and an easy experiment) if these were colocalized with a control marker to definitively show this.

3. Does every K13 complex have SAG1 foci directly above it? Do other non-GPI linked plasma membrane proteins show similar foci (eg. HP03)?

4. The authors should quantify plaque assays in Fig. 4C.

5. The authors state that "K13-knockdown parasite vacuoles shows inter-parasite spaces filled by single-membrane bound parasite cytoplasm. These extensions lacked an IMC but often contained recognizable parasite organelles."

- How was the lack of IMC assessed?

- Which organelles? - May be helpful to stain with rhoptries or other markers to strengthen this claim.

- Since the parasites are replicating, is the mitochondrion still in a classical lasso-shape? Or is it collapsed? (PMC7018656)

- Is it possible that this is enhanced residual body following replication rather than blebbing of cytosolic components?

6. Fig 6. - the vacuoles appear very different in 6B than 6C which has clear morphological differences that would be apparent in phase (also seen in S5). What percent of the vacuoles show the disruption of membrane integrity in 6C, 6D, and S5? (also Fig S5 is unclear. Are these just several examples at a single time point?)

Text changes

1. Line 29 - "are major unanswered questions" - would be clearer if the preceding were in a question format

2. Figure 1 - it would be helpful if the merged panels in each image were labeled

3. Figure 2A - TSC3 is referenced as Chern et al. but it should be Chen et al. 2015.

4. MCA2 (TGME49_243298) should be re-labeled as MCA3 based on previous publication (PMID: 34384491).

Reviewer #3:

Remarks to the Author:

Stable and ancient endocytic structures navigate the complex pellicle of apicomplexan parasites
Koreny et al 2022 Submitted to Nature Communications

Apicomplexan parasites, such as the malaria parasite and other such as *Toxoplasma* and *Cryptosporidium*, are major human and animal pathogens. A key characteristic of them is that they are intracellular and therefore need to gain nutrients, and interact with the "extracellular world", through their host cell. Furthermore, the importance of understanding this exchange is heightened in intracellular pathogens where it is necessary to ensure that drugs can gain access to the pathogens for treatment. In most cells, endocytosis is a key mechanism for this nutrient/environmental exchange, however, little is known about the mechanisms governing endocytosis in the Apicomplexa. This is a very important topic which this paper addresses, using the parasite *Toxoplasma* as a model.

In the paper the authors set out to determine where endocytosis occurs, to identify the major function of endocytosis and to investigate the mechanistic and origins of features of the process. The authors comprehensively address a sequence of questions using robust, highly detailed and well controlled experiments. As part of their investigations, they have developed a novel endocytosis assay that could be used more widely as a tool for these types of studies in the future. The methods are reported in a robust repeatable manner.

The authors identify the components of the micropore, the key role of the kelch-domain protein K13 and the link to endocytosis. Based on the phenotype of K13 knock-downs, the authors identify that the key function of endocytosis seems to be in membrane homeostasis rather than parasite nutrition – this is based on the fact that these knockdowns affect parasite membrane structure/organisation but does not affect parasite growth. The authors discuss the limitations of this conclusion, in detail, in the discussion but it is worded more definitively in the abstract. Finally, the authors place their findings in an evolutionary context and demonstrate the conservation of these structures.

The authors use approaches that are scientifically robust and that support their conclusions.

The presentation of the manuscript is generally good, although for the general reader there is a degree of complexity over gene/protein names especially in the initial parts of the results section. Minor comments

Line 36. The abstract implies that endocytosis is not essential to parasite nutrition. While the data points to the great importance in plasma membrane homeostasis, I don't think that the data, as it stands, fully rules out an important role in nutrition. The authors should remodel this sentence to reflect that. (It is well discussed in the discussion).

The authors should check through the manuscript and polish up/edit the language/grammar, just a couple of examples for instance:

Throughout (e.g. line 29, 34 etc), it is a better written style if "the Apicomplexa" rather than just "Apicomplexa" is used.

Line 202 *Toxoplasma* K13 complex shows conservation with the *Plasmodium* cytochrome but also "possesses" unique proteins.

Etc.

REVIEWER COMMENTS

Reviewer #1 (Remarks to the Author):

Koreny et al

NCOMMS-22-26335-T

This manuscript explores the relationship of the Kelch-13 centered complex and the IMC and pellicle of *T. gondii* that form fixed sites of endocytosis that are situated in gaps of the inner membrane complex. The core of the complex consists of Kelch 13 and AP2 adapter proteins that are associated with clathrin mediated endocytosis in other eukaryotes. Kelch13 and AP2 proteins have been previously characterized as components of the cytosome of *Plasmodium falciparum*, which is used by malaria parasites for hemoglobin uptake. Kelch-13, a core component of the complex, and other cytosome-associated proteins are implicated in artemisinin resistance.

The work uses multiple complementary techniques to understand the structure and organization of this complex, including 3D SIM super resolution microscopy and proteomics. The images are high quality and the overall model is convincing, but the manuscript suffers throughout because of qualitative rather than quantitative support for the observations.

The identification of a novel endocytic structure that is fixed and composed of both conserved and novel proteins is significant, particularly since the proteins appear to be conserved in Apicomplexa and essential for parasite survival.

1. The abstract could be rewritten to focus on the novel cell biology features of the findings. Yes, the endocytic complex proteins are implicated in artemisinin resistance in malaria, but this isn't particularly relevant for the current story that focuses upon *Toxoplasma* and identifies a conserved endocytic complex in Apicomplexa that is fixed and localized to the inner membrane complex.

Authors' response: We have modified the abstract to emphasise the more general conclusions of the study. However, we feel that it is also useful to retain mention of the importance of some of the molecules of this study (K13) and this structure to drug resistance in malaria as an outstanding medical problem.

2. Figure 1 and S1 show localization data of K13, 5 members of the AP2 associated family, and DrpC, a member of the dynamin family. The AP2 localize with K13 though Beta also also has other localization. They report DrpC is seen in 2 forms one similar to K13 and the other a conical punctum of smaller diameter consistent with constriction. How many images, replicates were used to define these localizations? Did they perform quantitation of localizations, quantify distances or perform statistical analyses?

Authors' response: Our conclusion that DrpC occurs in 2 forms (dilated ring and constricted punctum) is supported by 20 independently imaged structures (3 in Fig 1, 17 in Figure S2). There is, therefore, good support for these two different states. We do not make any conclusions of the relative frequency of these forms so statistical analyses are not relevant here. The locations of other AP2 proteins are also shown by multiple examples in Fig 1 and S1 and S2, and these locations are invariant in all of our studies.

3. Figure 2 shows localization of proteins identified by BioID with 5 bait proteins. IMC proteins were amongst the interacting proteins identified. Details of the criteria to fold change are presented as a heat map, but the details of the actual data are not provided, and the heat maps are very difficult to read with tiny fonts. From the methods provided, the analysis appears appropriate. The authors state the data have been deposited, but as presented in the manuscript the rigor and validity of the analysis are impossible to assess because of incomplete information. Further details of false and true positive data should be provided in a supplementary table.

Authors' response: We have used the BioID data as a discovery method for proteins in proximity to the K13 complex, but all of our conclusions are based on subsequent independent validations of colocalization of proteins by microscopy. Nevertheless, we do also recognise the value of these data for readers and their own independent enquiries. Thus, all of the raw mass spectrometry data has been made publicly available in the PRIDE database with identifier PXD034193, as given in the methods section. We have now also provided full details of the analysis outcomes, including the proteomics data containing the raw abundance values for 1420 Medium- and High-FDR-Confidence proteins identified across all the samples (Table S1), and the output from the statistical analysis of protein enrichments (fold change) by LIMMA linear modelling (Table S2) so that the heat map alone is not required for this information.

4. Figure 3 explores the positioning of the complex in relationship to the IMC. K13 localizes to longitudinal or transverse plate boundaries, and regions where IMC membranous markers are excluded, consistent with coordinated interruption of the IMC, with K13 providing an interface between plasma membrane and cytosol. Again, details of replicates, images analyzed and quantitation of these observations is not provided.

Authors' response: For all instances of K13 labelling with IMC markers multiple examples have been given. Additionally, we now provide a further 56 structure images in the Figs S4 and S5 showing the consistency of these observations over numerous replicates. Furthermore, we also cite Chern et al (2021) who independently located some of our micropore proteins to the sutures of Toxoplasma which provides further support to our observations.

5. Figure 4 Live fluorescent imaging reveals stable complexes and that the complexes appear to assemble during daughter cell formation prior to plasma membrane association with the IMC. Again no quantitation of these observations is provided. Plaque assays have no statistics or information about replicates.

Authors' response: Fig 4 shows 8 parasites with 16 developing daughters and all show nascent micropore complexes by the marker proteins used. This is consistent with our observations that we always see these proteins appearing during pre-cytokinesis development of the daughters. We have modified the results text to make this clear — "Invariably, we saw K13, KAE, AP-2 α , EPS15L and DrpC present as the characteristic complexes in these developing daughter IMCs (Fig 4B)."

The plaque assays were all performed three times independently giving the same result. This is now made clear in the figure legend and methods description.

6. Figure 5 presents a new endocytosis assay based upon Halo tagging of SAG1 that can be used to quantify external vs internalized SAG1. Using this assay, they show that K13 Knockdown is associated with decreased endocytosis of SAG1. Replicates of assays are not provided nor is any quantitation or statistics.

*Author response: This reviewer comment confuses us. Fig 5 shows statistical tests for all the endocytosis assays, standard deviations as error bars and P-values. The legend clearly states "Three biological replicates were used for all analyses; P-values are where $0.0001 < P \leq 0.001$, ***" and the details of the statistical tests given in the methods.*

7. Figure 6 shows that K13 KD doesn't affect parasite replication, but vacuoles are disorganized with impaired plasma membrane marker localization. Does K13 KD affect egress or subsequent invasion?

Authors' response: This is an interesting question that further addresses the state of the parasites after impairment of endocytosis, and we have addressed this by further experimentation. We have performed egress assays on K13-depleted parasites and show that there is indeed a strong impairment to egress (Fig 6E). This result is logical given the disorganisation of the parasite plasma membrane that is important for cytokinesis and motility, and it also provides a clear rationale for the plaque assay result where parasite

replication is otherwise not affected in the short term. Given this egress phenotype, it was not feasible to further investigate subsequent invasion of these affected cells.

Reviewer #2 (Remarks to the Author):

This study is centered around the TgK13 complex and its function in parasite endocytosis. The authors first use proximity labeling and high-resolution microscopy to identify and localize several components of the parasites endocytic machinery that likely corresponds to the micropore. They then develop a novel endocytosis assay based around the recycling of SAG1 and show that K13 is important for this dynamic process. Finally, the authors characterize the defects of K13-depleted parasites to demonstrate that the purpose of endocytosis in these parasites is to maintain plasma membrane homeostasis rather than uptake nutrition for growth. Overall, this is an important study that sheds light into a long-standing mystery with extensive techniques and robust evidence to support the conclusions. Specific comments are below.

Major comments

1. Overall, this is solid and important work. The identification of a group of proteins that are almost certainly the micropore is nicely done with excellent imaging and the use of proximity labelling with multiple bait proteins. In addition, the importance of the development of the endocytosis assay that assesses the recycling of SAG1 is important. Existing assays are tricky and hard to quantify and this advance provides a new method for assessing uptake via these proteins.

2. The authors never directly show that the protein machinery identified corresponds to the micropore. Could one of these proteins be directly localized to the micropore to firmly resolve this? (the reviewer acknowledges this may be technically challenging).

Authors' response: We initially tried to label HA-K13-tagged by immuno-EM without success. However, we are pleased to report that have created a new K13-tagged cell-line using a different epitope (V5 in tandem triplicate) and we have achieved clear and specific labelling, and that this shows that K13 is located at the micropore ultrastructure. This provides a very satisfying conclusion to this outstanding question, and we thank the reviewer for the extra encouragement to seek this answer.

3. The authors mention that many of these components contain cytoskeletal or protein-protein interaction domains that might make the micropore a more stable structure. Is K13 tethered to the cytoskeleton as assessed by detergent fractionation (as done for ISAP1 and its interactors in Chern et al.)

Authors' response: This question of the biophysics of K13 is beyond the immediate scope of questions that we address in this study, and is part of a subsequent study that we are performing focusing on the biochemistry of assembly and interactions of micropore proteins in detail.

4. The authors should better acknowledge the findings of previous work, in particular the overlap with Chern et al. which localized several of the same proteins and also showed localization to the IMC suture junctions (note that this does not diminish the findings of this paper which provides additional proteins and a much higher resolution of the components within the micropore and surrounding components of the IMC).

- Chern et al. previously reported the localization of ISAP1 and its partners - KAE (TGME49_272600), EPS15L (TGME49_227800), and CGAR (TGME49_297520) and showed colocalization to the sutures.

- DrpC was previously localized to discrete puncta in three separate papers (Melatti et al. 2019, Heredero-Bermejo et al. 2019, Chern et al. 2021)

- These findings should be included in the results (lines 168, 181-185), added to the previously localized

proteins in Fig 2A, and noted in Fig 3 (lines 227-237).

Authors' response: We have now included direct mention of the Chern et al report of ISAP1, EPS15L, CGAR, KAE and DrpC at the sutures in the results, and indeed this study reinforces our conclusions of a micropore at this location. The summaries of locations in Figs 2 and 3 are to the micropore and are therefore novel to our study. All of the studies mentioned related to DrpC (Melatti et al. 2019, Heredero-Bermejo et al. 2019, Chern et al. 2021) are cited.

5. Regarding epitope tagging endocytic machinery components

- a) Fig 1 - The authors should state which tags were used for each protein and whether they were N or C terminally tagged in the figure or in the figure legend. Same applies for other figures when needed or if changed for a particular experiment. This could also be accomplished with a supplementary table.
- b) It appears that N-terminal tagging was used in some cases (for K13 and KAE BioID at least). Is this because C-terminal tagging gave apparent mislocalization or inactivity of the fusion protein? – or was some feature of the protein that was avoided? If so, this should be discussed.

Authors' response: We now make it clear what reporter tags have been used and their termini of fusion in the figures/figure legends. It is correct that K13 and KAE did not tolerate C-terminal fusions and we make this clear in the methods now — “Reporters were C-terminally fused to their proteins of interest unless this was not tolerated (K13 and KAE-fusions could only be recovered when at the N-terminus).” Furthermore, we discuss that DrpC could be tagged at either end but that when tagged at the N-terminus (and with a non-native promoter) additional DrpC locations to the micropore were also seen. We limited our conclusions to the C-terminal fusion with native promoter.

6. How does K13 depletion affect the localization of its partners? Are they still present in foci at the IMC sutures? or is the complex disrupted and the proteins are then mislocalized?

Authors' response: These are interesting, but additional questions, that we are addressing in an exploration of the assembly of this structure as a separate detailed study.

7. Since the authors already have knockdown-inducible lines for other proteins, is it possible to show that KD of other components (or even a single other one) show similar plasma membrane homeostasis problems? Additional functional data would provide more compelling evidence of this paradigm shift for the function of the micropore in apicomplexans.

Authors' response: We have acted on this request and analysed the phenotypes of DrpC depletion also (Fig S9). We show that the same phenotypes develop as for K13 — no change in replication rate, but development of disordered vacuoles and large plasma membrane extensions seen by electron microscopy. This confirms that the plasma membrane homeostasis problems are a general phenotype of micropore disruption, not just that of K13. Furthermore, dynamin has a known functional role in endocytosis, so this result provides a direct link between the process and the phenotype. We agree that these additional functional data strengthen our conclusions.

8. Fig 6 “K13-depletion disrupts parasite order and integrity rather than replication rate” The authors state that replication rate is not affected - but then presumably shortly after the timepoint examined, the parasites die as no plaques are formed. Is this likely due to an incomplete knockdown at this stage? Discuss how protein levels and the phenotype ultimately affect viability?

Authors' response: We have now determined that egress is strongly affected by K13-depletion within the timeframe of this treatment, and this provides a clear rationale for the plaque assay result. Furthermore, we discuss that residual K13 protein cannot be discounted in our interpretation of phenotypes.

Minor comments

1. DrpC is not enriched very highly in multiple datasets compared to other components of the K13 complex.
- Melatti et al has shown DrpC associated with the mitochondria and Chern et al sees DrpC signal distinct from ISAP1. Is DrpC always K13 associated? I believe all of these were C-terminal tags thus unlikely to be the N-terminal tagging issue brought up in the paper?

Authors' response: The apparent association of DrpC with the mitochondrion is difficult to assess because Melatti et al did not have a micropore marker and their observation might reflect the chance proximity of the large mitochondrion with the micropores. Chern et al report some DrpC signal without ISAP1, however, in our ongoing studies of the micropore assembly we note that DrpC appears before ISAP1 in developing daughters, and it is possible that they have observed these early daughter cells. In our study we have used both immuno-fluorescence assays and live-cell markers, and we overwhelmingly observe DrpC exclusively with the K13 protein as we state in the discussion. It is true that in the BioID results the DrpC showed relatively less enrichment with the other micropore baits, however this result might be due to DrpC being the most distally located of the micropore proteins, or some steric hindrance or obstruction from other proteins that might shield it from reciprocal labelling. Interpretation of these relative signals is not straightforward, although we note that mitochondrial proteins do not appear in the list of BioID proximal proteins to DrpC.

2. Fig 2A, S3. The authors find several centrosome-like proteins and basal complex proteins in their BioID experiments. It would be stronger (and an easy experiment) if these were colocalized with a control marker to definitively show this.

Authors' response: These locations are putative and as these proteins do not inform on the K13 complex they didn't warrant further examination in the context of this study. We have modified the language to allow for uncertainty in the final location designations of these proteins.

3. Does every K13 complex have SAG1 foci directly above it? Do other non-GPI linked plasma membrane proteins show similar foci (eg. HP03)?

Authors' response: Upon reanalysing our 3D-SIM data and projection of the K13 signals from the side we found that the focus of SAG1 is actually a small penetration of the K13 ring in an extension consistent with the micropore membrane invagination. We now show 22 independent structures with SAG1 located with respect to K13 (Fig 3 and S6B). This imagery is close to the limits of resolution of this form of microscopy, and we do not think that it is suitable to definitively determine if some K13 rings lack a SAG1 extension, so this is not a question that we address. Nevertheless, the reproducibility of this observation is informative of how the micropore interacts with the plasma membrane. The HP03 marker was excellent in live cell imaging but did not fix well and we do not have access to 3D-SIM super resolution microscopy in a containment facility that would allow for live imaging at the necessary resolution to test if it also occurs in the membrane invagination of the micropore.

4. The authors should quantify plaque assays in Fig. 4C.

Initial response: The reviewer provides no rationale for this request or states what questions could be addressed by doing so. Indeed, none of our conclusions would be affected. We use the plaque assays to conclude that each of the proteins tested are independently necessary for some facet of normal micropore function where the result is binary — no difference in plaque size versus near or complete absence of plaque development. The interpretation of differing plaque sizes between different knockdown mutations is a very inexact science with assumptions of consistent rates of protein depletion, sensitivity of relative protein abundance, modes of action of mutation, and kinetics of plaque growth very difficult to validate in order for meaningful comparisons to be made. Therefore, we don't see any useful role for plaque quantification in the research questions that we've addressed or the conclusions that we present.

5. The authors state that “K13-knockdown parasite vacuoles shows inter-parasite spaces filled by single-membrane bound parasite cytoplasm. These extensions lacked an IMC but often contained recognizable parasite organelles.”

- How was the lack of IMC assessed?

- Which organelles? - May be helpful to stain with rhoptries or other markers to strengthen this claim.

Authors' response: the IMC is identifiable as a thick dark line bordering the parasites in the TEMs on account of it being comprised of two additional membranes to the plasma membrane plus a proteinaceous subpellicular network. Its absence is clear where only a single bounding membrane delineates the cytoplasmic extensions of the mutations. The organelles identifiable in the membrane extensions include the dense granules and mitochondria which are clearly identifiable by their distinct ultrastructure and staining patterns, and examples of these have been indicated in the figures now. Staining with markers is not trivial as a new, non-osmicated fixation would be required for antibody labelling.

- Since the parasites are replicating, is the mitochondrion still in a classical lasso-shape? Or is it collapsed? (PMC7018656)

Authors' response: Addressing any conformational changes to the mitochondria is not directly relevant to the primary questions of this study. Given that mitochondria are seen in the cytoplasmic extensions of the mutants, it is expected that mitochondrial morphology will be considerably disrupted but this may be a secondary effect of the disorder created by the loss of plasma membrane homeostasis.

- Is it possible that this is enhanced residual body following replication rather than blebbing of cytosolic components?

Authors' response: It is not clear what the reviewer means by 'enhanced residual body'. The residual body is still a relatively enigmatic structure, however it is known to be involved in maintaining the organised rosette formation of the parasite vacuoles and exchanging organelles and molecules between the precytokinetic cells. How this function could be 'enhanced' is not particularly clear. However, given that we see poorly organised vacuoles that fail to egress, it is much more likely that the phenotypes that we see represent a deterioration of function, rather than an enhancement.

6. Fig 6. – the vacuoles appear very different in 6B than 6C which has clear morphological differences that would be apparent in phase (also seen in S5). What percent of the vacuoles show the disruption of membrane integrity in 6C, 6D, and S5? (also Fig S5 is unclear. Are these just several examples at a single time point?)

Authors' response: The live cell imaging (Fig 6C and S8 [former S5]) using either GFP markers of the plasma membrane or DIC imaging does enable a superior view of the morphology of the cells and plasma membrane compared to fixed cells viewed by phase. This is likely due to some degree of membrane collapse or reorganisation during chemical fixation. Nevertheless, the fixed cells viewed with phase (Fig 6B) enable scoring of the vacuole arrangement (ordered versus disordered) and this is also the presentation that the field is most used to seeing. We think it is useful to show both for a fuller interpretation of the state of the cells. We have also now scored the percentage of vacuoles with disrupted membranes as requested and this is included in Fig 6. The legend for Fig S8 states that these images are further examples from Fig 6, and in Figure 6 it is stated that this is a single time point equivalent to the other treatments in this figure (i.e., 24 hours post invasion with a total of 72 hours of ATc treatment).

Text changes

1. Line 29 – “are major unanswered questions” - would be clearer if the preceding were in a question format

Authors' response: We have reworded this sentence as requested — “Where endocytosis occurs in these cells, how conserved this process is with other eukaryotes, and what the functions of endocytosis are across this phylum are major unanswered questions.”

2. Figure 1 - it would be helpful if the merged panels in each image were labelled

Authors' response: We have reorganised Fig 1 and the coloured signals shown in the merge are all clearly indicated.

3. Figure 2A - TSC3 is referenced as Chern et al. but it should be Chen et al. 2015.

Authors' response: This has been corrected although we note that the year of this publication is 2017.

4. MCA2 (TGME49_243298) should be re-labeled as MCA3 based on previous publication (PMID: 34384491).

Authors' response: This is slightly unfortunate because this Toxoplasma protein is orthologous to the Plasmodium MCA2. However, to avoid confusion with the Toxoplasma field we have used the precedent of MCA3 as suggested.

Reviewer #3 (Remarks to the Author):

Stable and ancient endocytic structures navigate the complex pellicle of apicomplexan parasites
Koreny et al 2022 Submitted to Nature Communications

Apicomplexan parasites, such as the malaria parasite and other such as Toxoplasma and Cryptosporidium, are major human and animal pathogens. A key characteristic of them is that they are intracellular and therefore need to gain nutrients, and interact with the “extracellular world”, through their host cell. Furthermore, the importance of understanding this exchange is heightened in intracellular pathogens where it is necessary to ensure that drugs can gain access to the pathogens for treatment. In most cells, endocytosis is a key mechanism for this nutrient/environmental exchange, however, little is known about the mechanisms governing endocytosis in the Apicomplexa. This is a very important topic which this paper addresses, using the parasite Toxoplasma as a model.

In the paper the authors set out to determine where endocytosis occurs, to identify the major function of endocytosis and to investigate the mechanistic and origins of features of the process.

The authors comprehensively address a sequence of questions using robust, highly detailed and well controlled experiments. As part of their investigations, they have developed a novel endocytosis assay that could be used more widely as a tool for these types of studies in the future. The methods are reported in a robust repeatable manner.

The authors identify the components of the micropore, the key role of the kelch-domain protein K13 and the link to endocytosis. Based on the phenotype of K13 knock-downs, the authors identify that the key function of endocytosis seems to be in membrane homeostasis rather than parasite nutrition – this is based on the fact that these knockdowns affect parasite membrane structure/organisation but does not affect parasite growth. The authors discuss the limitations of this conclusion, in detail, in the discussion but it is worded more definitively in the abstract. Finally, the authors place their findings in an evolutionary context and demonstrate the conservation of these structures.

The authors use approaches that are scientifically robust and that support their conclusions.

The presentation of the manuscript is generally good, although for the general reader there is a degree of complexity over gene/protein names especially in the initial parts of the results section.

Minor comments

Line 36. The abstract implies that endocytosis is not essential to parasite nutrition. While the data points to

the great importance in plasma membrane homeostasis, I don't think that the data, as it stands, fully rules out an important role in nutrition. The authors should remodel this sentence to reflect that. (It is well discussed in the discussion).

Authors' response: We agree with this point and have modified the Abstract sentence as follows — “We determine that a dominant function of endocytosis in Toxoplasma is plasma membrane homeostasis, rather than parasite nutrition, . . .”. In the discussion we consider that a nutritional role cannot be ruled out, but that it does not manifest in the primary phenotype upon endosymbiosis inhibition.

The authors should check through the manuscript and polish up/edit the language/grammar, just a couple of examples for instance:

Authors' response: we have done our best to correct language and grammar throughout.

Throughout (e.g. line 29, 34 etc), it is a better written style if “the Apicomplexa” rather than just “Apicomplexa” is used.

Authors' response: Apicomplexa is the name of the phylum and therefore we believe that our use is correct. It is often incorrectly used as a plural noun in place of 'apicomplexans' and we seek to avoid this.

Line 202 Toxoplasma K13 complex shows conservation with the Plasmodium cytostome but also “possesses” unique proteins.
Etc.

Authors' response: Thank you, corrected.

Reviewers' Comments:

Reviewer #1:

Remarks to the Author:

The authors have addressed my prior comments and supplied new data that strengthens their manuscript. The manuscript provides a thorough study that gives new insight into the Kelch 13 complex in *Toxoplasma gondii*.

Reviewer #2:

Remarks to the Author:

The authors have done a nice job with addressing the comments from the initial review. In particular the immunoelectron microscopy directly showing localization to the micropore is a great advance for this nice paper – congratulations!